# Functional abnormalities in the cerebello-thalamic pathways in a mouse model of DYT25 dystonia

**Hind Baba Aïssa[1†], Romain W Sala[1†], Elena Laura Georgescu Margarint[1†], Jimena Laura Frontera[1], Andrés Pablo Varani[1], Fabien Menardy[1], Assunta Pelosi[2,3,4], Denis Hervé[2,3,4], Clément Léna[1\*‡], Daniela Popa[1\*‡]**

[1]Neurophysiology of Brain Circuits Team, Institut de biologie de l'Ecole normale supérieure (IBENS), Ecole normale supérieure, CNRS, INSERM, PSL Research University, Paris, France; [2]Inserm UMR-S 1270, Paris, France; [3]Sorbonne Université, Sciences and Technology Faculty, Paris, France; [4]Institut du Fer à Moulin, Paris, France

**Abstract** Dystonia is often associated with functional alterations in the cerebello-thalamic pathways, which have been proposed to contribute to the disorder by propagating pathological firing patterns to the forebrain. Here, we examined the function of the cerebello-thalamic pathways in a model of DYT25 dystonia. DYT25 ($Gnal^{+/-}$) mice carry a heterozygous knockout mutation of the *Gnal* gene, which notably disrupts striatal function, and systemic or striatal administration of oxotremorine to these mice triggers dystonic symptoms. Our results reveal an increased cerebello-thalamic excitability in the presymptomatic state. Following the first dystonic episode, $Gnal^{+/-}$ mice in the asymptomatic state exhibit a further increase of the cerebello-thalamo-cortical excitability, which is maintained after θ-burst stimulations of the cerebellum. When administered in the symptomatic state induced by a cholinergic activation, these stimulations decreased the cerebello-thalamic excitability and reduced dystonic symptoms. In agreement with dystonia being a multiregional circuit disorder, our results suggest that the increased cerebello-thalamic excitability constitutes an early endophenotype, and that the cerebellum is a gateway for corrective therapies via the depression of cerebello-thalamic pathways.

**\*For correspondence:**
lena@biologie.ens.fr (CL);
dpopa@biologie.ens.fr (DP)

[†]These authors contributed equally to this work
[‡]These authors are co-directed to this work

**Competing interest:** The authors declare that no competing interests exist.

## Editor's evaluation

Baba Aissa et al. provide compelling evidence for a modulatory role of the cerebello-thalamo-striatal pathway in the pathology of DYT25 dystonia. Their results further suggest that cerebellar stimulation holds promise as a therapeutic intervention for treating dystonia.

## Introduction

Dystonia is a class of neurological disease whose symptomatology is characterized by involuntary movements and abnormal postures, caused by the co-contraction of antagonistic muscles (*Berardelli et al., 1998*; *Albanese et al., 2013*). The wide clinical spectrum of dystonia, whether in causes or clinical manifestations, has hampered so far the identification of a common pathophysiological mechanism underlying the disease (*Balint et al., 2018*). If early studies have primarily highlighted the role of the basal ganglia in the onset of the disease, more recent studies have revealed strong dysfunctions in the motor cortex, thalamus, and cerebellum in dystonic patients, questioning which structural impairments are primary or secondary causes of dystonia (*Simonyan, 2018*). Indeed, lesions in the basal

ganglia or in the cerebellum do not systematically induce a dystonic phenotype, and the onset of the disease is usually delayed following the lesions. Moreover, if the basal ganglia and cerebellum have been considered as therapeutic targets, approaches targeting these structures yielded inconsistent outcomes (*Oyama and Hattori, 2021*). These specificities led to the emergence of a new working hypothesis, that dystonia is a circuit disorder that has a diversity of triggering mechanisms but requires the interactions between several nodes of the motor network to reach a symptomatic state (*Lehéricy et al., 2013*; *Prudente et al., 2014*).

In line with the circuit disorder theory, studies of genetic forms of the disease have identified endophenotypes in the form of nonmotor symptoms, such as altered temporal discrimination threshold in nonmanifesting mutation carriers (*Hutchinson et al., 2013*), causing an inability of individuals to consider two subsequent stimuli as asynchronous if those are presented in a short amount of time. Such endophenotypes indicate an altered sensorimotor processing, which after amplification over time or through environmental interactions would lead to motor dysfunction. Anomalies in synaptic plasticity in the cortex and striatum have also been linked to dystonia in genetic forms of dystonia in rodent models (*Calabresi et al., 2016*). Cerebellar dysfunction preexisting to symptoms has also been observed in such models: nonmanifesting DYT6 mice exhibit an aberrant electrophysiological activity in the cerebellar cortex and nuclei, which is further disrupted in manifesting animals (*van der Heijden et al., 2021*). The functional connectivity in motor circuit may play an important role in the expression of symptoms. Indeed, DYT1 mutation carriers exhibit cerebello-thalamic disruptions, and the penetrance of DYT1 dystonia, as well as the severity of symptoms, is regulated by the structural integrity of cerebello-thalamo-cortical tracts (*Argyelan et al., 2009*). This observation has been reproduced in a mouse model of DYT1 (*Uluğ et al., 2011*), as well as a defect in cortico-striatal plasticity (*Yu-Taeger et al., 2020*), supporting the possibility of an inter-dependence of cerebello-thalamic and striatal dysfunctions. Thus, animal models of genetic forms of dystonia support the idea of a circuit disorder, but also offer an opportunity to investigate the network alterations at the level of the cerebellum, basal ganglia, motor cortex, and their reciprocal connections.

DYT25 is a recently identified genetic form of primary torsion dystonia, characterized as an autosomal-dominant adult-onset disorder (*Fuchs et al., 2013*; *Kumar et al., 2014*). It is caused by loss-of-function mutations of the *Gnal* gene encoding Gα(olf), a G-protein stimulating adenylate cyclase activity, mainly expressed in the olfactory bulb and striatum, with a sparse expression in Purkinje cells of the cerebellar cortex (*Belluscio et al., 1998*; *Vemula et al., 2013*). The genetic alterations discovered in DYT25 dystonic patients can be mimicked by heterozygous knockout mutation of the *Gnal* gene (*Gnal$^{+/-}$*) (*Pelosi et al., 2017*). In this model, cAMP production is reduced in the striatum, disrupting striatal functions, but the mice are devoid of dystonic symptoms in early adulthood. In agreement with a role of increased striatal cholinergic activity in dystonia (*Pisani et al., 2007*), dystonic symptoms are induced by injections of a muscarinic cholinergic agonist (oxotremorine M) administered either systemically or in the striatum, but not in the cerebellum, and were prevented by the muscarinic antagonist, trihexyphenidyl, a drug alleviating symptoms in many dystonic patients. This indicates that an increase in striatal cholinergic tone is critical to the onset of the disorder (*Pelosi et al., 2017*). This is consistent with a primary involvement of *Gnal* in striatal neurotransmission, despite its sparse expression in other brain regions (*Zhuang et al., 2000*; *Corvol et al., 2001*; *Hervé et al., 2001*; *Corvol et al., 2007*; *Vemula et al., 2013*).

Regardless of the heterogeneity in the function and expression of the genes involved in DYT1 and DYT25, these types of dystonia share many similarities. Strikingly, in animal models, similar alterations in cortico-striatal plasticity were observed in DYT1 and DYT25, suggesting common pathophysiological mechanisms (*Martella et al., 2021*). While the alteration of cerebellum or cerebello-thalamic tracts has been linked to the expression of DYT1 in patients and mouse models (*Argyelan et al., 2009*; *Uluğ et al., 2011*; *Fremont et al., 2017*), studies of DYT25 have remained focused on striatal alterations. However, an involvement of cerebello-thalamic tracts in the pathophysiology of DYT25 remains an open question. Moreover, cerebellar stimulations have been shown to be beneficial in idiopathic cervical dystonia (*Koch et al., 2014*; *Bradnam et al., 2016*), and genetic mouse models of dystonia allow a finer dissection of the effects induced by this type of approach (*van der Heijden et al., 2021*). In contrast to the DYT1 mouse, the DYT25 mouse model has the additional advantage of being pharmacologically inducible, allowing the study of the network's state in the presymptomatic and symptomatic states.

In this study, we therefore investigated the functional connectivity and plasticity of cerebello-thalamic tracts in the *Gnal* mouse model. We performed optogenetic stimulations in the cerebellar dentate nucleus (DN) and recorded activity in the ventrolateral thalamus (VAL, which projects to the motor cortex), centrolateral thalamus (CL, which projects to the striatum), primary motor cortex (M1), and dorsolateral striatum (DLS). We investigated the state of this network in the presymptomatic condition, symptomatic state, and asymptomatic state after the induction of the disease. Finally, in line with the therapeutic benefit of cerebellar stimulations in patients, we tested the effect of optogenetic θ-frequency stimulations of the DN on the plasticity of cerebello-thalamo-cortical tracts and on motor symptoms.

## Results

### Young adult *Gnal⁺ᐟ⁻* mice do not exhibit constitutive locomotor impairments

3–7-month-old *Gnal⁺ᐟ⁻* mice have been described as asymptomatic, without dystonic phenotype in control conditions (*Pelosi et al., 2017*). To further examine whether *Gnal⁺ᐟ⁻* mice exhibit constitutive motor deficit (taking into account gender), we performed a larger set of motor experiments, including vertical pole, horizontal bar, grid test, fixed-speed rotarod, gait test, and an open-field test (*Figure 1*, *Supplementary file 1a*).

We observed that the motor performance of *Gnal⁺ᐟ⁻* mice was not impaired in the vertical pole test (*Figure 1A*, *Supplementary file 1a*), horizontal bar test (*Figure 1B*, *Supplementary file 1a*), and grid test (*Figure 1C*, *Supplementary file 1a*). Females exhibited better performances in the horizontal bar test, regardless of their genotype.

Motor coordination was also examined in a fixed-speed rotarod test. We did not find significant differences between *Gnal⁺ᐟ⁻* and WT mice (*Figure 1D*, *Supplementary file 1a*), nor when comparing males and females for each speed step (*Figure 1E*, *Supplementary file 1a*).

In the gait test, we found no significant differences in gait width, alternation coefficient, linear movement, sigma, or length of stride (*Figure 1F*, *Supplementary file 1a*).

Finally, in spontaneous locomotion in the open field, we observed no significant differences between genotype and gender neither in median instantaneous speed nor in distance traveled (*Figure 1G, J and K*, *Supplementary file 1a*). Additionally, we included an analysis of the thigmotaxis and time spent exploring the center of the open field and observed no significant differences between genotypes or genders in our mice (*Figure 1H, I*, *Supplementary file 1a*). In conclusion, motor activity and motor coordination are not impaired in young 3–7-month-old *Gnal⁺ᐟ⁻* mice; furthermore, no significant differences were observed between male and female *Gnal⁺ᐟ⁻* mice compared to their WT littermates, allowing us to merge genders in the following physiological experiments.

### Thalamo-cortical alteration in *Gnal⁺ᐟ⁻* mice in presymptomatic and symptomatic states

To assess the functional connectivity of the ascending pathway connecting the cerebellum with the thalamus and motor cortex, we performed extracellular recordings of the VAL and CL thalamic nuclei, as well as M1 in awake freely moving young adult mice of both genotypes (*Figure 2*, *Figure 2—figure supplement 1*, *Supplementary file 1b*). Mice were recorded at the age of 3–7 months, at which *Gnal⁺ᐟ⁻* mice are asymptomatic, before the onset of an abnormality of motor coordination (*Pelosi et al., 2017*). VAL, CL, and M1 neurons presented no differences in firing rate between WT and *Gnal⁺ᐟ⁻* mice in basal conditions (*Figure 2—figure supplement 2A*, *Supplementary file 1k and l*).

To induce a dystonic-like phenotype in *Gnal⁺ᐟ⁻* mice (*Pelosi et al., 2017*), we then injected oxotremorine M (0.1 mg/kg, i.p.), a nonselective cholinergic agonist in both WT and *Gnal⁺ᐟ⁻* mice. As described previously, this treatment induces in both genotypes a cholinergic shock, which translates into excessive salivation and lacrimation, as well as a persistent immobility. It also induces in *Gnal⁺ᐟ⁻* mice abnormal dystonic-like movements and postures. Oxotremorine-treated mice, whether WT or *Gnal⁺ᐟ⁻*, presented a significant decrease in the firing rate of VAL and M1 (*Figure 2—figure supplement 2A*, *Supplementary file 1k and l*). While oxotremorine did not modify the firing properties of CL neurons in WT mice, it decreased the firing rate of CL neurons in *Gnal⁺ᐟ⁻* mice (*Figure 2—figure supplement 2A*, *Supplementary file 1k and l*). We then investigated the effect of a first exposure to oxotremorine

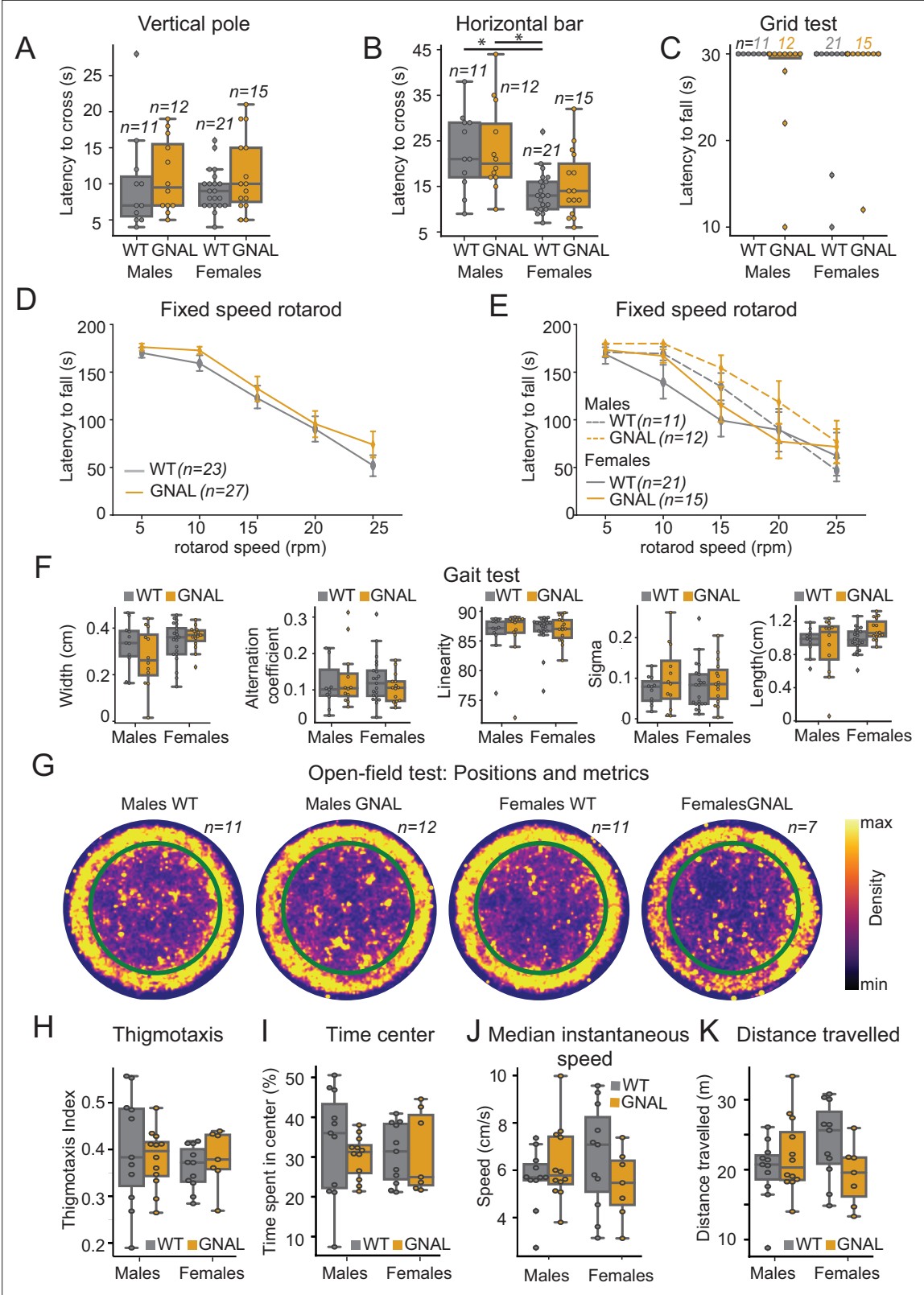

**Figure 1.** Young male and female *Gnal⁺/⁻* mice do not display motor coordination impairments. (**A**) Latency to climb down the vertical pole in male and female *Gnal⁺/⁻* (GNAL) and wildtype (WT) mice. (**B**) Latency to cross the horizontal bar. (**C**) Latency to fall during the grid test (30 s cutoff). (**D**) Latency to fall during the fixed-speed rotarod test separated by genotype, and (**E**) by gender and genotype. (**F**) Gait width, alternation coefficient, movement linearity, sigma, and stride length during the gait test. (**G**) Two-dimensional histograms showing the density of position of mice during open-field

*Figure 1 continued on next page*

Figure 1 continued

sessions, separated by gender and genotype. Thigmotaxis index (**H**), percentage of time spent in center (**I**), median instantaneous speed (**J**), and total distance traveled (**K**) during the open-field test. Reported statistics are the comparison of groups separated by gender and genotype using a Kruskal–Wallis test, followed by a Dunn's post-hoc test corrected using Holm–Sidak method. *p<0.05.

on the regularity of firing patterns in the thalamus. We observed a significant decrease in $CV_{isi}$ of VAL neurons in both WT and $Gnal^{+/-}$ mice (*Figure 2—figure supplement 3A*, *Supplementary file 1o and p*). While oxotremorine did not modify the $CV_{isi}$ of CL neurons in WT mice, it was decreased for CL neurons in $Gnal^{+/-}$ mice (*Figure 2—figure supplement 3A*, *Supplementary file 1o and p*), suggesting an increased regularity of firing.

Overall, these results show no thalamic impairments in $Gnal^{+/-}$ mice in the presymptomatic state compared to WT mice but reveal differences between genotypes following the induction of dystonic-like state using oxotremorine injection. Oxotremorine by itself affects VAL-M1 activity, while a sensitivity of the CL was only found in $Gnal^{+/-}$ mice compared to WT mice.

## First exposure to cholinergic agonist induces long-lasting changes in cerebello-thalamic excitability

The DN is the main output of the cerebellum towards the thalamus and is known to project both to the VAL and CL (*Ichinohe et al., 2000*; *Teune et al., 2000*). To probe the activity of these cerebello-thalamic pathways, we hence paired recordings of these structures with low-frequency, low-intensity stimulations of the DN in $Gnal^{+/-}$ and WT mice not only in the presymptomatic state (saline-injected naive mice) and symptomatic (oxotremorine-injected mice) states, but also in an asymptomatic state (saline-injected mice at least 2 days after oxotremorine injection) (*Figure 2C*, *Figure 2—figure supplement 2B and C*).

In naive saline-treated mice, we found a significant increase in firing rate for the VAL, CL thalamus and M1 neurons during 100 ms DN stimulations in WT and $Gnal^{+/-}$ mice (*Figure 2—figure supplement 2B and C*). Responses in the VAL were significantly larger in $Gnal^{+/-}$ than in WT mice, suggesting an increased responsiveness of this thalamo-cortical pathway in the presymptomatic state. We verified the absence of plastic changes of the responses in VAL, CL, and M1 induced by our low-frequency stimulation of the DN by showing the absence of significant difference between the average response to the first half of the stimulations ('early') and the second half of the stimulations ('late,' *Figure 2—figure supplement 2B and D*, *Supplementary file 1n*).

Then, we examined the effect of acute exposure to oxotremorine on cerebello-thalamo-cortical projections. The naive saline-treated mice were thereafter injected with oxotremorine and subjected again to low-frequency stimulations ('Oxo' condition, *Figure 2A and C*). We did not observe any significant difference in response to DN stimulation in the VAL, CL, and M1 in comparison to naive condition, for both genotypes (*Supplementary file 1c and d*). This suggests that the first exposure to oxotremorine does not cause significant short-term changes in the cerebellar drive of thalamic nuclei and motor cortex.

In order to investigate a potential long-term effect of the exposure to oxotremorine on cerebello-thalamo-cortical sensitivity, the mice were subjected once more to low-frequency, low-intensity opto-genetic stimulations of the DN in saline conditions, 2 days after their first exposition to oxotremorine ('Post-Oxo,' asymptomatic condition, *Figure 2A*). While the acute exposure to oxotremorine yielded slightly increased responses in VAL neurons and decreased responses in M1 neurons in WT mice, $Gnal^{+/-}$ mice in the asymptomatic state presented a significantly larger response to DN stimulations in all recorded structures compared to the changes induced by oxotremorine in WT animals (*Figure 2C*, *Supplementary file 1c and d*), indicating an increased excitability of the cerebello-thalamic pathways in asymptomatic $Gnal^{+/-}$ mice following the first dystonia induction.

Overall, these results show that a single exposure of $Gnal^{+/-}$ mice to oxotremorine induces, in the cerebello-thalamic pathways, long-lasting functional alterations that are more pronounced than the mild alterations observed in the presymptomatic state.

## Cerebello-thalamic plasticity is altered in asymptomatic $Gnal^{+/-}$ mice

The long-lasting increase after oxotremorine treatment in $Gnal^{+/-}$ mice of the excitability in the cerebello-thalamic connections could be associated with an alteration of their plasticity. To test this

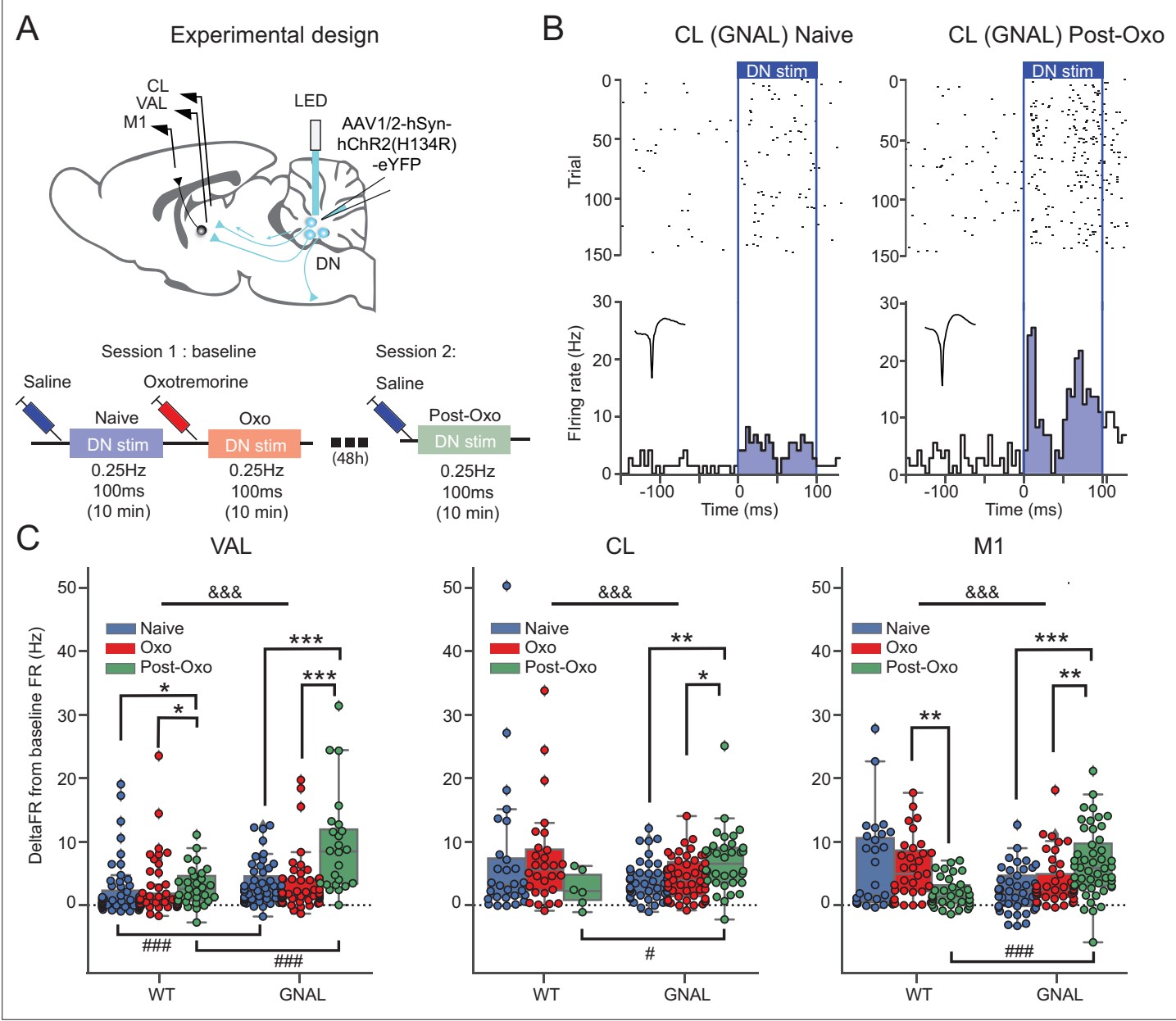

**Figure 2.** Exposure to oxotremorine causes a long-lasting increase in thalamic and cortical response to dentate nucleus (DN) stimulation in *Gnal+/−* mice. (**A**) Schematics describing the experimental design of thalamo-cortical recordings coupled with DN stimulation (top) and experimental timeline (bottom). (**B**) Examples of peristimulus time histogram (PSTH) and corresponding raster plot, centered on the onset of the cerebellar stimulation, of centrolateral thalamus (CL) neurons from the same recording site in a *Gnal+/−* mouse under saline condition, before being exposed to oxotremorine ('naive,' left) and 48–72 hr after being exposed to oxotremorine ('Post-Oxo,' right). Inset represents the average waveforms of the recorded neuron. (**C**) Distributions of responses to DN stimulations in naive condition (blue), under the acute effect of oxotremorine (red) and in saline post-oxo condition (green). Two-way ANOVA, with state and genotype as factors, followed by a Dunn's post-hoc test corrected using Holm–Sidak method to compare states within genotype. *p<0.05, **p<0.01, ***p<0.001; &&&p<0.001 for ANOVA interaction term between states and genotype.

The online version of this article includes the following figure supplement(s) for figure 2:

**Figure supplement 1.** Electrophysiology recordings in thalamo-cortical and cerebello-striatal networks.

**Figure supplement 2.** Effects of initial exposure to oxotremorine and dentate nucleus (DN) stimulation on the thalamo-cortical network in wildtype (WT) and *Gnal+/−* mice.

**Figure supplement 3.** Effects of initial exposure to oxotremorine on the regularity of thalamic firing patterns in wildtype (WT) and *Gnal+/−* mice.

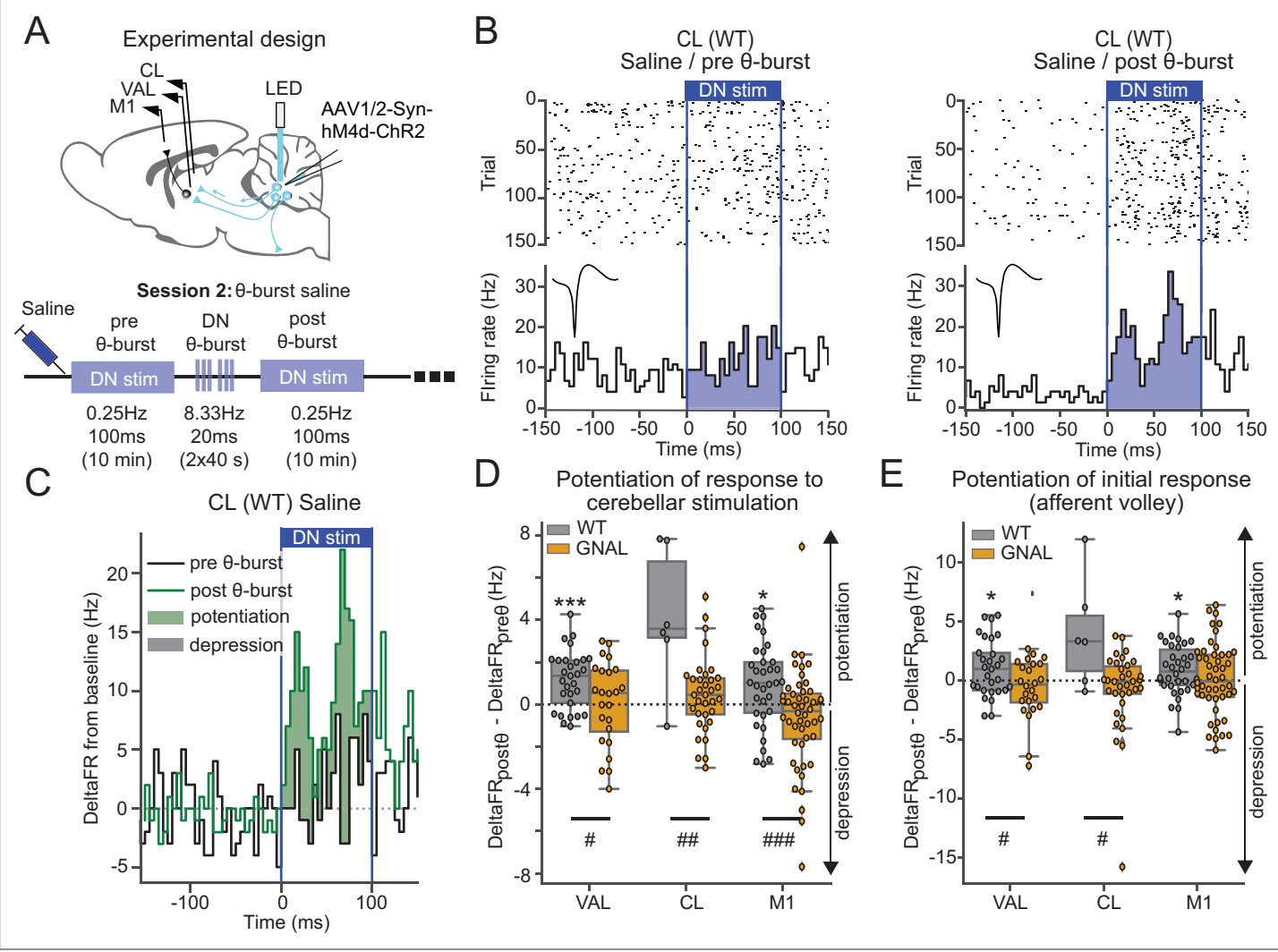

**Figure 3.** Asymptomatic *Gnal+/-* mice display an abnormal cerebello-thalamic plasticity induced by dentate nucleus (DN) $\theta$-bursts. (**A**) Schematics describing the experimental design of thalamo-cortical recordings coupled with dentate nucleus-centrolateral thalamus (DN-CL) stimulation (top) and experimental timeline (bottom). All the mice received oxotremorine (0.1 mg/kg) in session 1, which took place 2–3 days before. (**B**) Example of peristimulus time histogram (PSTH) and corresponding raster plot, centered on the onset of the cerebellar stimulation, of a CL neuron from a wildtype (WT) mouse under saline condition, before $\theta$-burst (left) and after $\theta$-burst (right). Inset represents the average waveform of the neuron. (**C**) Overlay of PSTHs for the neuron shown in panel (**B**); the difference between the histograms is filled to highlight the potentiation or depression of the responses. (**D**) Impact of $\theta$-burst stimulations administered in saline condition on the response to 100 ms DN stimulations. (**E**) Impact of $\theta$-burst stimulations administered in saline condition on the afferent volley in response to 100 ms DN stimulation. Wilcoxon test for paired samples *$p<0.05$, **$p<0.01$, ***$p<0.001$. Mann–Whitney test for independent samples #$p<0.05$, ##$p<0.01$, ###$p<0.001$ difference between genotypes.

possibility, we performed optogenetic θ-burst stimulations of the DN and examined the changes in thalamic and cortical responses to cerebellar stimulation by subsequently applying low-frequency DN stimulations (*Figure 3*).

In saline-treated WT mice, the comparison of the increase in firing rate induced by DN stimulations before and after θ-burst stimulations (*Figure 3B and C*) revealed a potentiation of the response in VAL and M1 neurons to the low-frequency stimulation (*Figure 3D*, *Supplementary file 1e*). On the other hand, in *Gnal+/-* mice, θ-burst stimulations failed to induce a significant potentiation in the responses of VAL, CL, or M1 neurons to DN stimulations, yielding significantly smaller θ-burst-induced changes in response to DN stimulations compared to WT mice (*Figure 3D*, *Supplementary file 1e*). We verified that these effects did not result from the recruitment of other brain structures over the duration of the 100 ms stimulation by quantifying the change in response in a short (10 ms wide) initial window corresponding to the putative afferent volley. This analysis reduced the spikes counts considered and

thus decreased the signal/noise ratio of detection of potentiation. However, a reduction of θ-burst effect was observed in *Gnal*[+/-] mice compared to WT mice in VAL and CL neurons. In M1, the significant θ-burst-induced potentiation was only observed in WT but not *Gnal*[+/-] mice (*Figure 3E*). Hence, in the saline condition, θ-burst stimulations of the DN induced a potentiated response for WT mice in motor thalamo-cortical pathways, whereas this potentiation was absent in *Gnal*[+/-] mice, indicating an impaired plasticity of cerebello-thalamic pathways in the asymptomatic state.

## Effect of cerebellar stimulation on the cerebello-striatal pathway

The cerebello-CL-striatal pathway (*Ichinohe et al., 2000*; *Bostan and Strick, 2010*; *Chen et al., 2014*; *Gornati et al., 2018*; *Xiao et al., 2018*) has been shown to play a central role in certain forms of dystonia (*Chen et al., 2014*). We therefore further investigated the functionality of the cerebello-striatal pathway by recording awake freely moving mice where DN neurons were retrogradely infected from the CL with a ChR2-expressing AAV virus (*Figure 4A*). With the use of an optrode, we recorded the activity of DN neurons while performing optogenetic stimulations in the DN and recorded simultaneously extracellular activity from the DLS.

Using 100 ms low-frequency DN stimulations, we observed in both *Gnal*[+/-] and WT mice an increased firing rate in DN neurons consistent with the existence of a population of CL-projecting DN neurons (*Figure 4B and C*, *Supplementary file 1f*). Application of a θ-burst protocol in these mice (*Figure 4—figure supplement 1*, *Supplementary file 1q*) failed to induce a global potentiation or depression of DN responses to subsequent low-frequency 100 ms stimulations (*Figure 4D*), suggesting that θ-burst stimulations in the DN do not result in a long-lasting change of excitability in the stimulated neurons. Finally, we examined the impact of oxotremorine injections on DN discharge and found, as in other recorded brain regions, a significant decrease in firing rate in DN neurons under oxotremorine conditions in both *Gnal*[+/-] and WT mice (*Figure 4—figure supplement 2A*, *Supplementary file 1r*). However, contrarily to the impact of oxotremorine injections on CL discharge (*Figure 2—figure supplement 2A*), we found no such difference between the genotypes in the DN (*Figure 4—figure supplement 2A*, *Supplementary file 1r*). Interestingly, while dystonia has been associated with irregular cerebellar firing in other models of dystonia (*LeDoux and Lorden, 1998*; *Chen et al., 2014*; *Fremont et al., 2014*; *Fremont et al., 2017*; *van der Heijden et al., 2021*), the firing irregularity (as measured by the coefficient of variation of the interspike interval, $CV_{isi}$) was not significantly modified under oxotremorine for WT and *Gnal*[+/-] mice, although DN neurons in *Gnal*[+/-] mice exhibited a stronger shift toward regular firing discharge under oxotremorine compared to WT mice (*Figure 4—figure supplement 2B*, *Supplementary file 1r*).

The impact of these DN stimulations was then examined in the DLS. DLS neurons exhibited a bimodal distribution of firing rate (*Figure 4E*), allowing to separate slow- and fast-spiking neurons corresponding to putative medium spiny neurons and interneurons (*Her et al., 2016*). Neurons from both populations in the DLS displayed significant departure from baseline firing during DN optogenetic stimulations in both WT and *Gnal*[+/-] mice, but there was no significant increase in the firing rate of the population over the whole stimulation interval (*Figure 4F*, *Supplementary file 1f*). Θ-burst stimulations of the DN also produced a rhythmic entrainment of the DLS fast-spiking populations (*Figure 4—figure supplement 1B and C*) consistent with a striatal entrainment through CL-projecting DN neurons.

## Aberrant striatal plasticity in asymptomatic *Gnal*[+/-] mice

We then examined the long-lasting effect of θ-burst DN-CL stimulations on DLS responses in the saline condition (*Figure 4G–J*). In WT animals, DLS fast-spiking neurons exhibited increased responses to low-frequency 100 ms DN stimulations after θ-burst protocol, not only when taking into account the whole duration of the DN stimulation (*Figure 4H and I*), but also when focusing on the initial response (*Figure 4J*). This suggests that θ-burst DN stimulations increase the recruitment of DLS fast-spiking units through an oligo-synaptic pathway. In contrast, such change in response to DN low-frequency 100 ms stimulations was absent in *Gnal*[+/-] mice in either type of units in the DLS following θ-burst stimulations. This observed difference between genotypes is consistent with the impact of θ-burst stimulations in the CL neurons in saline condition (*Figure 3D and E*) and thus indicates an impairment in cerebello-thalamo-striatal plasticity in asymptomatic ('post-Oxo') *Gnal*[+/-] mice.

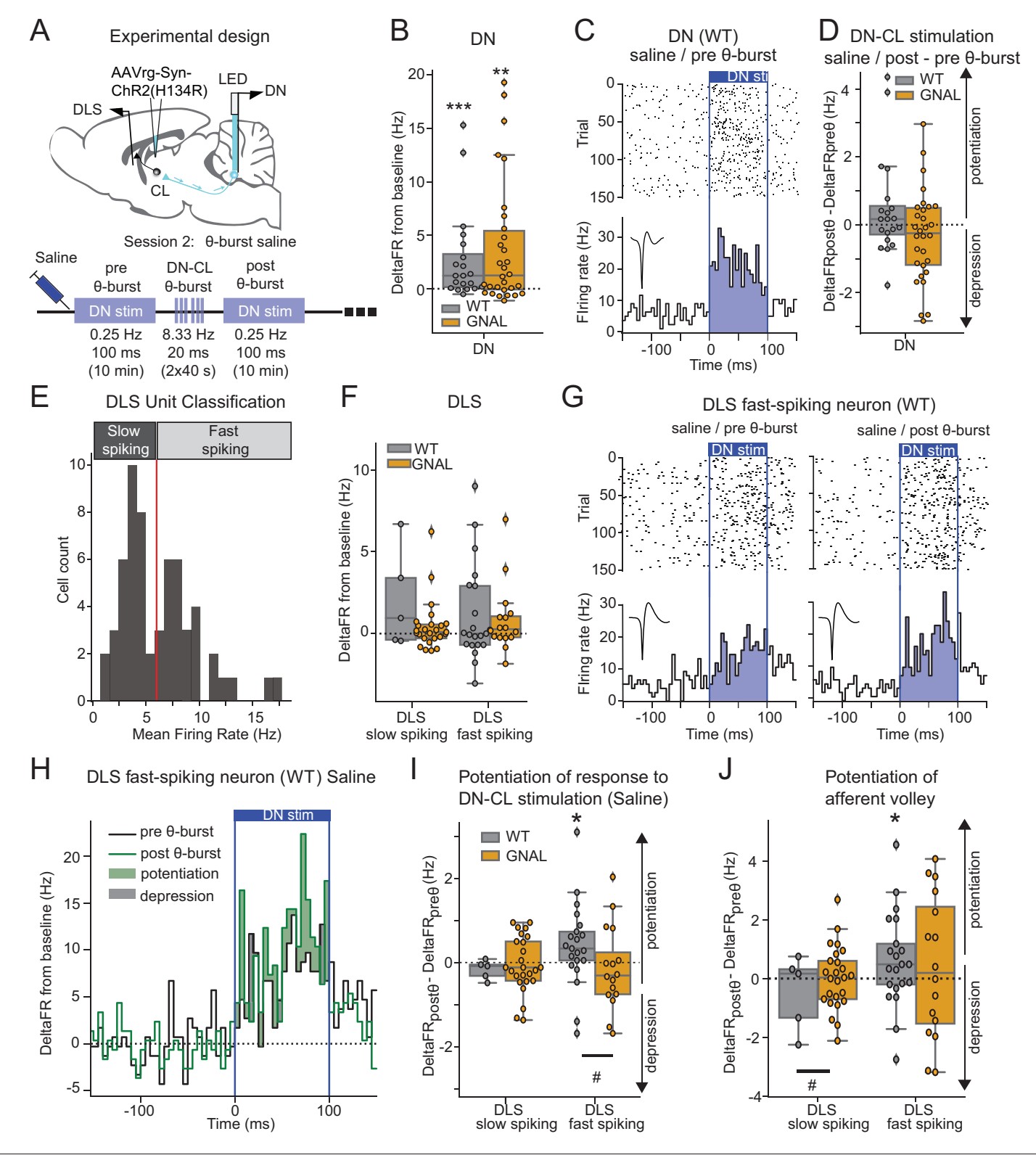

**Figure 4.** Dentate nucleus-centrolateral thalamus (DN-CL) $\theta$-bursts potentiate the response of fast-spiking dorsolateral striatum (DLS) neurons to DN stimulation in WT wildtype (mice), but not in *Gnal*[+/-] mice. (**A**) Schematics describing the experimental design of cerebello-striatal recordings coupled with DN-CL stimulation (top) and experimental timeline (bottom). (**B**) Distributions of response amplitude in the DN to optogenetic stimulation in saline condition, before $\theta$-bursts. (**C**) Example of peristimulus time histogram (PSTH) and corresponding raster plot, centered on the onset of the

*Figure 4 continued on next page*

*Figure 4 continued*

cerebellar stimulation, of a DN neuron from a WT mouse under saline condition, before $\theta$-bursts. Inset represents the average waveform of the neuron. (**D**) Impact of $\theta$-burst stimulations administered in saline condition on the amplitude of response to 100 ms DN-CL stimulation. (**E**) Illustration of the criteria used to classify DLS neurons as slow spiking and fast spiking based on their mean firing rate using a threshold of 6 Hz (red line). (**F**) Distributions of striatal responses to DN-CL stimulation in saline condition, before $\theta$-burst. (**G**) Example of a PSTH and corresponding raster plot, centered on the onset of the DN-CL stimulation, of a DLS fast-spiking neuron from a WT mouse under saline condition, before $\theta$-burst (left) and after $\theta$-burst (right). Inset represents the average waveform of the neuron. (**H**) Overlay of the PSTHs from panel (**G**); the difference between the histograms is filled to highlight the potentiation or depression of responses. (**I**) Impact of $\theta$-burst stimulations administered in saline condition on the response to 100 ms DN-CL stimulation. (**J**) Impact of $\theta$-burst stimulations administered in saline condition on the afferent volley in response to 100 ms DN-CL stimulation. Wilcoxon test for paired samples *p<0.05, **p<0.01, ***p<0.001. Mann–Whitney test for independent samples #p<0.05, ##p<0.01, ###p<0.001 for differences between genotypes.

The online version of this article includes the following figure supplement(s) for figure 4:

**Figure supplement 1.** Dentate nucleus-centrolateral thalamus (DN-CL) $\theta$-bursts elicit excitation in DN neurons and dorsolateral striatum (DLS) fast-spiking neurons.

**Figure supplement 2.** Effect of oxotremorine on the firing of the dentate nucleus (DN).

## Aberrant cerebello-thalamic plasticity in *Gnal*<sup>+/-</sup> mice in the symptomatic state

Since cerebellar θ-burst transcranial stimulation protocols have been shown to produce symptomatic relief in human motor disorders (*Koch et al., 2014*; *Bradnam et al., 2016*), we then examined the effects on motor circuits and behavior of θ-burst stimulations applied during oxotremorine-induced dystonia in our mice (*Figure 5*). Following oxotremorine administration, the increase in firing rate in response to low-frequency 100 ms DN stimulations in VAL, CL, and M1 neurons was stronger in *Gnal*<sup>+/-</sup> mice than WT mice (*Figure 5B*), as also found in the asymptomatic condition (*Figure 2C*).

In WT mice, θ-burst stimulations under oxotremorine did not induce lasting changes in the response of VAL and M1 neurons to low-frequency 100 ms DN stimulations, neither when taking into account the whole duration of stimulation (*Figure 5E*, *Supplementary file 1i*), nor when looking only at the initial response ('afferent volley,' *Figure 5F*, *Supplementary file 1i*). In contrast, θ-burst stimulations in *Gnal*<sup>+/-</sup> mice induced a lasting depression of the response to low-frequency cerebellar stimulations in VAL, CL, and M1 for the whole duration of the stimulation (*Figure 5C–E*, *Supplementary file 1i*). This depression was also observed in the afferent volley (*Figure 5F*, *Supplementary file 1i*). Thus, while cerebellar drive of the thalamus and cortex is enhanced in asymptomatic *Gnal*<sup>+/-</sup> mice, θ-bursts applied in symptomatic (but not asymptomatic, *Figure 3E and F*) *Gnal*<sup>+/-</sup> mice induce a decrease in the entrainment of the thalamus and M1 by the cerebellum.

## Cerebellar θ-burst stimulations reduce dystonic symptoms in *Gnal*<sup>+/-</sup> mice

We then evaluated the effect of optogenetic θ-burst stimulations on the motor state of *Gnal*<sup>+/-</sup> mice compared to WT mice (*Figure 5G–I*). In *Gnal*<sup>+/-</sup> mice, oxotremorine consistently induced abnormal postures such as the extension of hind limbs from the body axis for >10 s, sustained hunched posture with little movements, slow walking with increased hind limb gait, yielding maximal dystonia scores, while only mild motor signs were observed in WT mice (*Figure 5H*, *Supplementary file 1j*). These observations are consistent with the previous findings in *Gnal*<sup>+/-</sup> mice (*Pelosi et al., 2017*) and indicate that the surgical interventions (electrode and optical fiber implantation, AAV infections) did not impact the development of dystonic-type motor abnormalities.

DN optogenetic θ-burst stimulations decreased the abnormal motor state in *Gnal*<sup>+/-</sup> mice following oxotremorine injection (*Figure 5H*). To confirm these observations, we also evaluated the activity of mice by measuring the percentage of time spent moving in an open field ('active wakefulness') (*Georgescu et al., 2018*). In the saline-treated WT mice, DN optogenetic θ-burst stimulations had no significant effect on the active wake time. Similarly, in the asymptomatic *Gnal*<sup>+/-</sup> mice, θ-burst stimulations did not significantly change the percentage of time spent in active wake (*Figure 5I*, *Supplementary file 1j*). In contrast, in the oxotremorine condition, the average active wake time decreased in both WT and *Gnal*<sup>+/-</sup> mice compared to saline, the effect being more pronounced in the mutant mice (*Figure 5I*). One session of DN optogenetic θ-burst stimulation was sufficient to increase the time

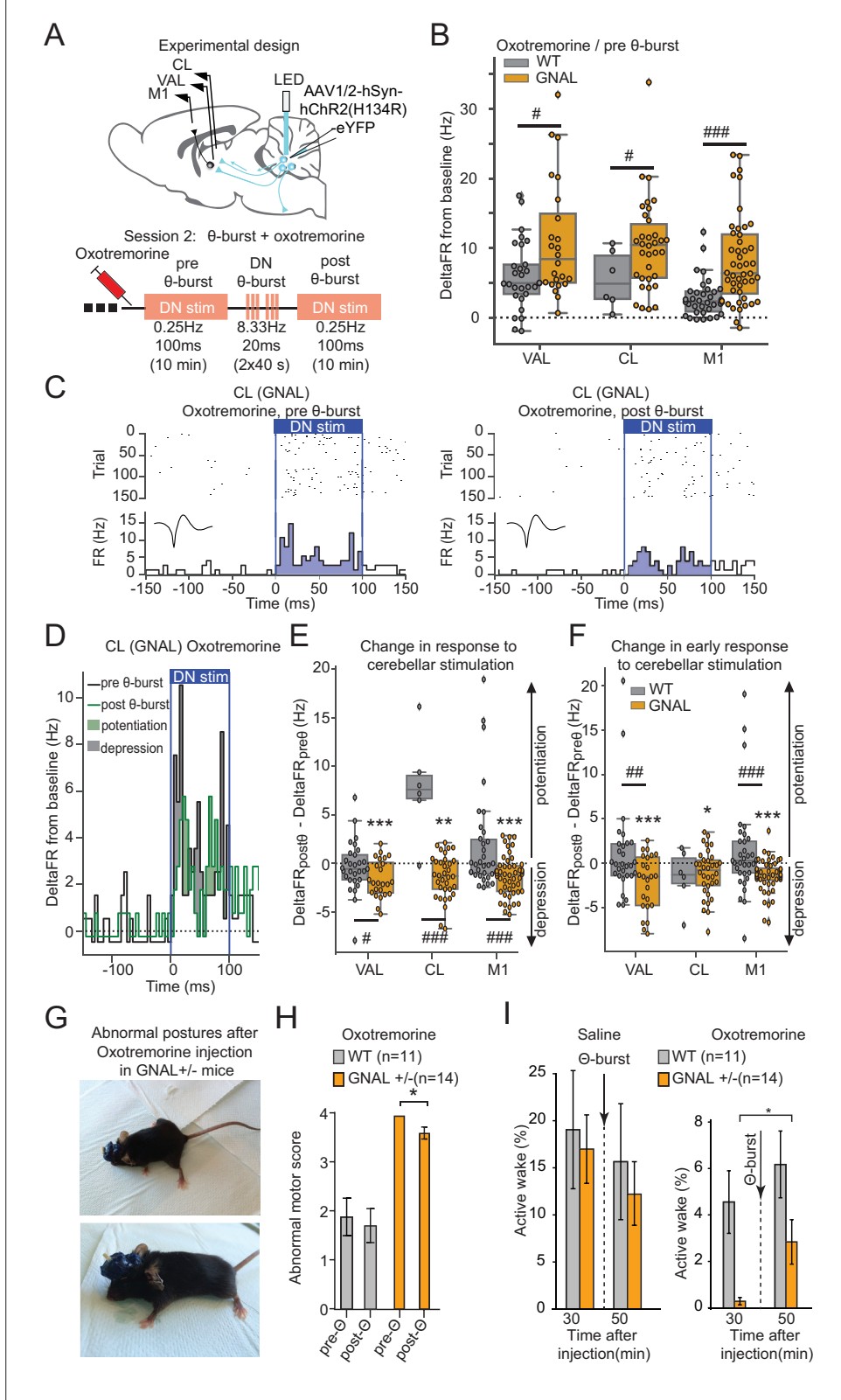

**Figure 5.** Dentate nucleus (DN) $\theta$-bursts administered to symptomatic *Gnal*[+/-] mice induce a decreased response to DN stimulation and decrease motor symptoms of dystonia. (**A**) Schematics describing the experimental design of thalamo-cortical recordings coupled with DN stimulation (top) and experimental timeline (bottom). (**B**) Distributions of responses to DN stimulation in oxotremorine condition, before $\theta$-burst. (**C**) Example peristimulus

*Figure 5 continued on next page*

*Figure 5 continued*

time histogram (PSTH) and corresponding raster plot, centered on the onset of the cerebellar stimulation of a centrolateral thalamus (CL) neuron from a *Gnal$^{+/-}$* mouse under oxotremorine condition, before $\theta$-burst (left) and after $\theta$-burst (right). Inset represents the average waveform of the neuron. (**D**) Overlay of the PSTHs from panel (**C**); the difference between the histograms is filled to highlight the potentiation or depression of the responses. (**E**) Impact of $\theta$-burst stimulations administered in oxotremorine condition on the response to +/-ms DN stimulation. (**F**) Impact of $\theta$-burst stimulations administered in oxotremorine condition on the afferent volley in response to 100 ms DN stimulation. Wilcoxon test for paired samples *p<0.05, **p<0.01, ***p<0.001; Mann–Whitney test for independent samples #p<0.05, ##p<0.01, ###p<0.001 difference between genotypes. (**G**) Examples of dystonic postures in *Gnal$^{+/-}$* mice following oxotremorine administration. (**H**) Average dystonia scores in *Gnal$^{+/-}$* and wildtype (WT) mice following oxotremorine administration before and after DN $\theta$-burst stimulations. (**I**) Change of average active wake percentage after one session of DN $\theta$-burst stimulations in *Gnal$^{+/-}$* and WT mice. Wilcoxon test *p<0.05 difference between pre- and post-$\theta$-burst stimulations.

spent in active wake in *Gnal$^{+/-}$* mice (**Figure 5I**, **Supplementary file 1j**). Overall, these results indicate that DN optogenetic θ-burst stimulations decrease the dystonic phenotype in symptomatic *Gnal$^{+/-}$* mice regarding both the abnormal motor scores and active wake deficit.

## Discussion

Our results show that young adult *Gnal$^{+/-}$* mice in the presymptomatic state, that is, without overt motor impairment, already exhibit altered cerebello-thalamic functional connectivity. After the first dystonic episode triggered by a cholinergic agonist, mutant animals, once returned to an asymptomatic state, exhibit further functional alterations of the cerebello-thalamic pathways, with a marked increase in cerebello-thalamic excitability and loss of potentiation of these functional connections by cerebellar θ-burst stimulations, which are also reflected downstream in the motor cortex and striatum. Finally, cerebellar θ-burst stimulations in the symptomatic state induce a functional depression of the cerebello-thalamic pathway absent from control mice, and this effect is accompanied by a reduction of the motor symptoms. This shows that in *Gnal$^{+/-}$* mice before the first dystonia episode the cerebello-thalamic tract exhibits an increased excitability that is amplified after the first episode and can thus be seen as a functional endophenotype. Indeed, the plasticity induced in this pathway via cerebellar stimulations reducing the cerebello-thalamic excitability exerts an effective therapeutic potential.

### DYT25 penetrance and dystonia endophenotypes

Mutations in *Gnal* (DYT25) were first identified in 2012 (*Fuchs et al., 2013*) as a cause of primary torsion dystonia. DYT25 is known to be an adult-onset dystonia, with an age at onset ranging from 7 to 63 years (average of about 30 years), and, although the onset occurs mainly in the neck (about 80%), dystonia can affect various body parts, with generalization in 10% of the cases (*Fuchs et al., 2013*; *Miao et al., 2013*; *Vemula et al., 2013*; *Dufke et al., 2014*; *Kumar et al., 2014*; *Ziegan et al., 2014*; *Saunders-Pullman et al., 2014*; *Carecchio et al., 2016*; *Dos Santos et al., 2016*). Interestingly, the immunohistochemistry study in rats of Gα(olf), the protein encoded by *Gnal*, reveals labeling in the olfactory bulb, striatum, substantia nigra, and cerebellar cortex (*Vemula et al., 2013*). In the striatum, Gα(olf) is closely associated with the G protein subunits, Gγ7 and Gβ2, and the type 5 adenylyl cyclase (AC5) to form a multi-molecular complex critical for cAMP production (*Hervé, 2011*; *Xie et al., 2015*) and mutations of Gγ7 and AC5 cause motor disorders, including dystonia, in animals and humans. Homozygous Gγ7 knockout mice, which have a severe impairment of Gα(olf) signaling in the striatum, display age-dependent dystonia (*Sasaki et al., 2013*). In addition, loss-of-function mutations of ADCY5, the human gene encoding AC5, are responsible for chorea with dystonia (*Carapito et al., 2015*), whereas gain-of-function mutations cause dyskinesia with facial myokymia (*Chen et al., 2015*). Thus, the disruption of the striatal transduction pathway involving Gα(olf) likely plays a central role in the development of dystonia.

In the mouse striatum, expression of Gα(olf) highly increases between postnatal day 7 (P7) and P14 (*Iwamoto et al., 2004*), a period during which motor skills of pups are developing intensely. Moreover, while juvenile rats (P14) displayed intense staining in cell bodies of striatal cholinergic neurons and medium spiny neurons, adult rats displayed a more diffused staining in the striatum, with a weaker staining in cholinergic neurons. Similarly, juvenile rats displayed a staining of the soma and dendrites

of Purkinje cells, while it was restricted to their soma in adult rats. These shifts in expression suggest that Gα(olf) plays a role in shaping neuronal networks and that the mutation of *Gnal* may cause an abnormal neuronal circuit development. DYT25 is inherited in an autosomal-dominant manner, but the penetrance of the disease is only partial (*Fuchs et al., 2013*; *Vemula et al., 2013*), and the higher penetrance and earlier childhood onset in carriers of the homozygous mutation (*Masuho et al., 2016*) is consistent with a contribution of *Gnal* to the maturation of the motor system.

While old (>1 year) *Gnal*^+/- adult mice exhibit motor impairments and occasional dystonic-like posture (*Pelosi et al., 2017*), our results in young adult *Gnal*^+/- mice failed to evidence motor deficits either in simple activities (locomotion and strength test) or in skilled movements (vertical rod, bar test, rotarod). They indeed provide a model of the presymptomatic stage of dystonia. Identifying presymptomatic alterations may provide insights into the mechanism by which the network activity will shift toward the pathological state (mediational endophenotypes), although these alterations may simply be a side effect of the mutation unrelated to the disease (*Hutchinson et al., 2013*). In humans, several physiological markers were proposed as endophenotypes of primary dystonia. The tactile temporal discrimination threshold (TDT) is increased in different types of isolated dystonia, nonmanifesting DYT1-mutation carriers, and unaffected relatives of both familial and sporadic adult-onset dystonia patients, and the somatotopic representation of fingers in the S1 cortex was disorganized on both sides in unilaterally affected dystonic patients (*Meunier et al., 2001*). Inter-hemispheric inhibition, as assessed by dual-site paired-pulse TMS, is reduced in asymptomatic individuals but with a family history of dystonia (*Bäumer et al., 2016*). A critical issue is that these subclinical physiological alterations may not be specific to dystonia: for example, TDT was found abnormal in Parkinson's disease (*Conte et al., 2016*) and psychogenic dystonia (*Morgante et al., 2011*). Some neuroimaging studies also demonstrated a pattern of hypermetabolism of the basal ganglia, cerebellum, and supplementary motor area linked with DYT1 and DYT6 dystonia even in nonmanifesting gene carriers (*Eidelberg et al., 1998*; *Trost et al., 2002*). While cerebello-thalamic fiber tract integrity in these dystonias is diversely affected in all gene carriers, nonmanifesting subjects exhibit a combination of rather preserved cerebello-thalamic tract and reduced downstream thalamus-motor cortex tract, suggesting that symptom manifestation due to disrupted cerebello-thalamic communication may be prevented by reduced thalamo-cortical interactions, as seen in nonmanifesting carriers (*Argyelan et al., 2009*). Although our data did not reveal significant increases in neuronal activity in naive *Gnal*^+/- mice that would compare to the hypermetabolism found in DYT1 or DYT6 patients, we found an anomalously high responsiveness of thalamus VAL neurons to cerebellar stimulations; since we found no difference in DN responsiveness to optogenetic modulations between genotypes, it is likely that this difference results at least in part from an increased synaptic transmission or postsynaptic excitability. The origin of this alteration is unclear but could result from structural defects or homeostatic regulations (i.e., compensatory mechanisms) of the cerebello-thalamic pathway.

## Cerebello-thalamic network after dystonia induction in *Gnal*^+/- mice

Oxotremorine injections affected both control and *Gnal*^+/- mice, but induced dystonic postures in the mutant mice. While these injections induced an overall decrease in firing rate in the DN, VAL, and M1 in both genotypes, the reduction in firing rate in the CL was only visible in mutant mice. Since only *Gnal*^+/- mice expressed a strong dystonic phenotype, an oxotremorine-induced reduction of firing may correspond to a rather nonspecific effect of the cholinergic stimulation in the DN, VAL, and M1 while the specific change in the CL of *Gnal*^+/- most likely reflects abnormal activity in the thalamo-striatal pathway in relation with dystonic symptoms. We did not observe a difference between genotypes in the reduction of discharge or irregularity of DN neurons under oxotremorine in *Gnal*^+/- mice, nor a decreased excitability to optogenetic stimulations. This contrasts with the observations from murine models of DYT1, DYT12, DYT6 dystonia or from dystonic *dt* rats, where increased irregularity of neuronal discharge is observed in the cerebellar nuclei (*LeDoux and Lorden, 1998*; *Chen et al., 2014*; *Fremont et al., 2014*; *Fremont et al., 2017*; *van der Heijden et al., 2021*), as also seen in dystonia caused by cerebellar infusion with kainate (e.g., *Pizoli et al., 2002*). This increased discharge irregularity could therefore be a landmark of dystonia with a primary cerebellar contribution (*LeDoux et al., 1993*; *Fremont et al., 2017*). In contrast to these rodent models, dystonia in *Gnal*^+/- mice might mainly result from the action of oxotremorine in the striatum of *Gnal*^+/- mice (*Pelosi et al., 2017*) and hence exhibit less fewer abnormalities in the cerebellar nuclei discharge.

The first dystonia episode in *Gnal*$^{+/-}$ mice was followed by strong and lasting increases in cerebello-thalamic excitability and stronger responses to DN stimulations in the motor cortex in the asymptomatic state. Such direct assessment of the cerebello-thalamic functional connectivity cannot be performed in patients. However, integrated measures of connectivity such as cerebellar brain inhibition (CBI, *Ugawa et al., 1995*), which measures – by subtraction – the tonic excitatory effect exerted by the cerebellar nuclei over the motor cortex, or the covariance of metabolic activity in the brain motor circuits, which reflects inter-regional functional connectivity, have been performed in several cohorts of dystonic patients (for the most part focal hand or cervical dystonia). These studies rather point toward a reduction of the entrainment of the cerebellum over downstream regions (*Brighina et al., 2009*; *Koch et al., 2014*; *Filip et al., 2017*; *DeSimone et al., 2019*; *Porcacchia et al., 2019*; *Panyakaew et al., 2020*; but see *Kita et al., 2021*). So far, the significance of these alterations is unclear: while a reduction of CBI was found in focal hand dystonia, it was either reduced or normal in cervical dystonia (*Brighina et al., 2009*; *Koch et al., 2014*; *Porcacchia et al., 2019*). In a rodent model of DYT12 dystonia, the cerebello-thalamic tract propagates pathological activities toward the striatum (*Chen et al., 2014*), suggesting that a decreased cerebello-thalamic functionality could exert a protective effect. Abnormal involvement of the cerebellum in the motor circuits has also been linked to the presence of tremor in dystonia (*DeSimone et al., 2019*), for which accumulating evidence correlates this feature with cerebellar dysfunction (*Antelmi et al., 2017*; *Martino et al., 2020*). Indeed, dynamic causal modeling of brain activity from patients with dystonic tremor suggests a central contribution of the cerebellum and cerebello-thalamic pathway in this form of tremor (*Nieuwhof et al., 2022*). Overall, the functional state of cerebello-thalamic pathway may thus depend on the type of dystonia. Our results suggest that in a type of dystonia with a primary striatal dysfunction (*Pelosi et al., 2017*) the hyperexcitability of the cerebello-thalamic tract could reveal an increased susceptibility to dystonia, already present in presymptomatic *Gnal*$^{+/-}$ mice and further potentiated after the first dystonic episode; a decrease in functional connectivity could however correspond to a long-term adaptation of the cerebello-forebrain pathways, which did not take place in the timescale of our experiments.

## Effects of θ-burst stimulations in *Gnal*$^{+/-}$ mice

Our results showed a differential sensitivity of the ascending cerebello-thalamic pathways to θ-burst stimulations of the DN in *Gnal*$^{+/-}$ and WT mice. In WT mice, these stimulations induce a lasting potentiation of responses in the downstream structures. They do not affect the response in the DN, indicating that their effect occurs downstream either as an increase in thalamic excitability or as a potentiation of cerebello-thalamic synapses (*Aumann et al., 2000*). In contrast, in *Gnal*$^{+/-}$ mice, the θ-burst stimulations failed to evoke a potentiation in saline condition. Since in these experiments the effect of DN stimulations on downstream structures has already been potentiated following the first dystonic episode, the failure to potentiate the responses to DN in the asymptomatic state could result from a saturation of the potentiation. In contrast, θ-burst stimulations administered in the presence of oxotremorine were followed by a depression of the cerebello-thalamic responses only in *Gnal*$^{+/-}$ mice, consistent with an abnormal cerebello-thalamic state or sensitivity to the cholinergic excitation.

Impaired plasticity is a landmark feature at cortico-striatal synapses in hyperkinetic disorders (*Calabresi et al., 2016*) and also found for cortical plasticity in transcranial experiments in focal dystonia patients (*Quartarone et al., 2005*). So far, the effect of the cerebellum on plasticity at the level of the thalamus, cortex, or striatum in dystonia is not well understood. In healthy subjects, cerebellar transcranial θ-burst protocols alter the cerebral cortex responsiveness, intra-cortical inhibition, and propensity to develop sensorimotor plasticity in the cortex (*Koch et al., 2008*; *Popa et al., 2010*; *Popa et al., 2013*); these protocols generally fail, in focal dystonia, to restore the influence of the cerebellum on the cerebral cortex (*Hubsch et al., 2013*; *Koch et al., 2014*; *Bologna et al., 2016*; *Bradnam et al., 2016*; *Porcacchia et al., 2019*). Transcranial cerebellar stimulations are thought to recruit Purkinje cells and thus affect the cerebellar nuclei, but the site where these transcranial protocols elicit plastic changes is not established. Interestingly, daily sessions of θ-burst protocols have been found to mildly improve the condition of dystonic patients (*Koch et al., 2014*; *Bradnam et al., 2016*). Alternatively, deep brain (high frequency) stimulations in patients indicate an improvement of dystonia, and notably tremor, when performed in the cerebello-thalamic pathway (*Cury et al., 2017*; *Coenen et al., 2020*). Stimulating the cerebellar nuclei has also been proven beneficial in

the treatment of secondary dystonia associated with palsy (*Sokal et al., 2015*). In all these studies, as in our experiments, the size effects of the treatment remained modest, but their impact on the life quality of patients may be noteworthy (e.g., *Bradnam et al., 2016*). Finally, repeated sessions of θ-burst stimulation might potentiate the effect (*Meunier et al., 2015*). In light of our results and consistent with the 'protective' effect of depressed cerebello-thalamo-cortical connections (*Argyelan et al., 2009*), the depression of the cerebello-thalamic pathway could thus be a mechanism by which cerebellar manipulations improve the condition of patients.

### *Gnal*$^{+/-}$ mice display an abnormal striatal plasticity

We observed short-latency responses in the DLS following stimulations of CL-projecting DN neurons, consistent with the disynaptic cerebello-striatal connection (*Bostan et al., 2013*) primarily relayed through the intralaminar thalamus (*Ichinohe et al., 2000*; *Chen et al., 2014*; *Gornati et al., 2018*) and targeting both interneurons and medium spiny neurons (*Xiao et al., 2018*). Additionally, we found that DN θ-stimulations induce a potentiation of the response of fast-spiking striatal neurons in control mice. This potentiation could result from the potentiation at the level of the CL; however, CL striatal afferents have been shown to be less prone to synaptic plasticity in medium spiny striatal neurons (*Ellender et al., 2013*), so the difference of effect of DN θ-burst between slow- and fast-spiking could also result to the recruitment of synaptic plasticity at the striatal level. In contrast to control mice, θ-burst stimulations did not induce cerebello-thalamo-striatal plasticity in asymptomatic *Gnal*$^{+/-}$ mice.

Defects in striatal plasticity have been found in murine models of primary dystonia DYT25 and DYT1: cortico-striatal synapses in DYT1 mice could undergo LTP but not LTD, and a previously-potentiated synapse could not be depotentiated (*Maltese et al., 2014*). Similarly, a loss of cortico-striatal LTD was observed in a DYT25 rat model. This loss of LTD could be rescued by blocking adenosine A2A receptors (*Yu-Taeger et al., 2020*) or by negatively modulating mGluR5 receptors (*Martella et al., 2021*). Interestingly, the striatal inputs from the CL have been shown to shift the cortico-striatal plasticity in favor of LTP (*Chen et al., 2014*); therefore, the increased excitability of the DN-CL pathway after the first dystonic episode could participate to maintain potentiated cortico-striatal synapses. However, the cortico-striatal plasticity is controlled by many parameters, including notably the striatal acetylcholine (*Deffains and Bergman, 2015*), which is altered in dystonia. Further work clarifying the control of cortico-striatal plasticity by thalamic inputs (*Mendes et al., 2020*) is needed to understand how the cerebello-thalamo-striatal projections contribute to the impairments of cortico-striatal plasticity in dystonia.

In conclusion, our study investigates an original dystonia model that mimics the genetic alterations discovered in patients suffering from the recently identified DYT25 dystonia and demonstrates the benefits of a model with a pharmacological switch between presymptomatic, symptomatic, and asymptomatic states. Although the striatum is likely the primary origin of functional alterations (*Pelosi et al., 2017*), our study reveals the presence of early abnormalities in cerebello-thalamic pathways in *Gnal*$^{+/-}$ mice and thus supports the view that dystonia is a motor network disorder. Furthermore, our results suggest that identifying cerebellar stimulation patterns that maximize the depression of the cerebello-thalamic pathway in patients could help improve therapeutic interventions.

## Materials and methods

### Animals

Experiments were performed in accordance with the guidelines of the European Community Council Directives. *Gnal*$^{+/-}$ mice were mated with C57BL/6J mice in order to obtain male and female *Gnal*$^{+/-}$ and WT littermates. Animals (males and females *Gnal*$^{+/-}$ and WT aged 3–7 months old) were kept at a constant room temperature and humidity on 12 hr light/dark cycle and with ad libitum access to water and food. All the motor control experiments were performed in males and females, and recordings were performed in freely moving mice.

### Open-field activity

Mice were placed in a circular arena made of polyvinyl chloride with 38 cm diameter and 15 cm height (Noldus, Netherlands) and video-recorded from above. Each mouse was placed in the open field for 5 min with the experimenter out of its view. The center of gravity of the mice was tracked using an

algorithm programmed in Python 3.5 and the OpenCV 4 library. Each frame obtained from the open field's videos was analyzed according to the following process: open-field area was selected and extracted in order to be transformed into a grayscale image. A binary threshold was then applied to this grayscale image to differentiate the mouse from the white background. To reduce the noise induced by the recording cable or by particles potentially present in the open field, a bilateral filter and a Gaussian blur were sequentially applied since those components are supposed to have a higher spatial frequency compared to the mouse. Finally, the OpenCV implementation of the Canny algorithm was applied to detect the contours of the mouse; the position of the mouse was computed as the mouse's center of mass. The distance traveled by the mouse between two consecutive frames was calculated as the variation of the position of the mouse center point multiplied by a scale factor to allow the conversion from pixel unit to centimeters. The total distance traveled was obtained by summing the previously calculated distances throughout the entire open-field session. The speed was computed as the variation of position of center points on two consecutive frames divided by the time between these frames (the inverse of the number of frames per second). This speed was then averaged by creating sliding windows of 1 s. After each session, fecal boles were removed and the floor was wiped clean with a damp cloth and dried after the passing of each mouse.

### Horizontal bar test

Motor coordination and balance were estimated with the horizontal bar test, which consists of a linear horizontal bar extended between two supports (length: 90 cm; diameter: 1.5 cm; height: 40 cm from a padded surface). The mouse is placed on one of the sides of the bar and released when all four paws gripped it. The mouse must cross the bar from one side to the other, and latencies to cross the bar are measured in a single trial session with a 3 min cutoff period.

### Vertical pole test

Motor coordination was estimated with the vertical pole test. The vertical pole (51 cm in length and 1.5 cm in diameter) was wrapped with white masking tape to provide a firm grip. Mice were placed heads up near the top of the pole and released when all four paws gripped the pole. The bottom section of the pole was fixated to its home-cage with the bedding present but without littermates. When placed on the pole, animals naturally tilt downward and climb down the length of the pole to reach their home cage. The time taken before going down to the home-cage with all four paws was recorded. A 20 s habituation was performed before placing mice at the top of the pole. The test was given in a single trial session with a 3 min cutoff period.

### Gait test

Motor coordination was also evaluated by analyzing gait patterns. Mouse footprints were used to estimate foot opening angles and hind base width, reflecting the extent of muscle loosening. The mice crossed an illuminated alley, 70 cm in length, 8 cm in width, and 16 cm in height, before entering a dark box at the end. Their hind paws were coated with nontoxic, water-soluble ink, and the alley floor was covered with sheets of white paper. To obtain clearly visible footprints, at least three trials were conducted. The footprints were then scanned and examined with the Dvrtk software (Jean-Luc Vonesch, IGBMC). The stride length was measured with hind base width formed by the distance between the right and left hind paws. The footprint pattern generated was scored for five parameters (*Simon et al., 2004*). Step length, the average distance of forward movement between alternate steps, is defined as the distance of travel divided by the number of steps. Sigma, describing the regularity of step length, is defined as the standard variation of all right-right and left-left step distance. Gait width, the average lateral distance between opposite left and right steps, is determined by measuring the perpendicular distance of a given step to a line connecting its opposite preceding and succeeding steps. Alternation coefficient, describing the uniformity of step alternation, is calculated by the mean of the absolute value of 0.5 minus the ratio of right-left distance to right-right step distance for every left-right step pair. Linearity, average change in angle between consecutive right-right steps is calculated by drawing a line perpendicular to direction of travel, starting at first right footprint. After determining angle between this perpendicular line and each subsequent right footprint, differences in angle were estimated between each consecutive step pair, and the average of absolute values of all angles was calculated.

## Grid test

The grid test is performed to measure the animal strength. It consists of placing the animal on a grid that tilts from a horizontal position of 0–180°. The time elapsed until the animal drops is recorded. The time limit for this experiment is 30 s. In those cases where the mice climbed up to the top of the grid, a maximum latency of 30 s was applied.

## Fixed-speed rotarod

Motor coordination, postural stability, and fatigue were estimated with the rotarod (mouse rotarod, Ugo Basile). The mice were placed on top of the plastic roller facing away from the experimenter's view and tested at constant speeds (5, 10, 15, 20, and 25 rpm). Latencies before falling were measured for up to 3 min in a single trial session.

## Surgery

Two surgeries were performed. During the first surgery, AAV2/1.hSyn.ChR2(H134R)-eYFP.WPRE.hGH (700 nl) was injected into the dentate nucleus of cerebellum (DN) of the $Gnal^{+/-}$ and WT mice (−6 mm AP, ±2.3 mm ML, −2.4 mm depth from dura). After 3 weeks, implantation surgery was performed. For both surgeries, the mice were anesthetized either with a mixture of ketamine/xylazine or a mixture of isoflurane and $O_2$ (3% for induction, 1.7% for maintenance). Injections with buprenorphine (0.05 mg/kg, s.c.) were performed to control pain, and core temperature (37°C) was kept with a heating pad. The mice were fixed in a stereotaxic apparatus (David Kopf Instruments, USA). After a local midline lidocaine injection s.c. (2%, 1 ml), a medial incision was performed, exposing the skull. Small craniotomies were drilled above the recording sites and above the optic fiber location (above the virus injection site), and then the electrodes were stereotaxically lowered inside the brain. This procedure allowed us, in one experimental set, to record in the left motor cortex (M1) (AP +2 mm and −2 mm ML from the Bregma, DV = −0.5 mm depth from the dura), ventrolateral thalamus (VAL) (−1.34 mm AP, ML = −1.00 mm, and DV = −3.4 mm depth from the dura), and centrolateral thalamus (CL) (AP at −1.58 mm, ML = −0.8 mm, DV = −3.00 mm depth from the dura). On a second experimental set, we recorded in the left dorsomedial striatum (DLS, −6 mm AP,±2.3 mm ML, −2.4 mm depth from dura) and dentate nucleus of cerebellum (DN, −6 mm AP, ±2.3 mm ML, −2.4 mm depth from dura). The mice were implanted with bundles of extracellular electrodes for each recording site. The ground wire was placed on the surface of cerebellum. Super Bond cement (Dental Adhesive Resin Cement, Sun Medical CO, Japan) was applied on the surface of the skull to strengthen the connection between the bone and the cement. The cannulas and ground wire were then fixed with dental cement (Pi-Ku-Plast HP 36, Bredent GmbH, Germany). The bundles of eight electrodes were made in-house by folding and twisting the nichrome wire with a 0.005-inch diameter (Kanthal RO-800) (*Menardy et al., 2019*). The bundles were placed inside guide cannulas (8–10 mm length and 0.16–0.18 mm inner diameter, Coopers Needle Works Limited, UK) glued (Loctite universal glue) to an electrode interface board (EIB-16; Neuralynx, Bozeman, MT) with one wire for each channel and four channels for each brain region (M1, CL, VAL), extending 0.5 mm below the tube tip. Wires were then fixed to the EIB with gold pins (Neuralynx), and then the EIB was secured in place by dental cement. A gold solution (cyanure-free gold solution, Sifco, France) was used for gold plating, and the impedance of each electrode was set to 200–500 kΩ.

## Manipulations of cerebellar output

Because the cerebellar nuclei send projections in the contralateral thalamus that then connects with the M1 (*Teune et al., 2000*), optogenetic stimulations were performed in the contralateral cerebellar DN (left M1 and right DN). Light-induced excitation of the cerebellar-projection neurons was elicited by using an LED driver (Mightex Systems) through optical fibers radiating blue light (470 nm) unilaterally implanted into the deep cerebellar nuclei (light intensity of 1.5 mW/mm²). Optogenetic stimulations of the DN were either 100 ms, 0.25 Hz, or θ-burst stimulation, 20 ms, 8.33 Hz, applied for 2 × 40 s with a 2 min pause in between and were performed before and after triggering the dystonic attacks by an oxotremorine methiodide (oxotremorine M) intraperitoneal injection.

## Electrophysiological recordings

The recordings began after at least 3 days of recovery and were performed on awake freely moving mice using a 16-channel acquisition system with a sampling rate of 25 kHz (Tucker-Davis Technology System 3, Tucker-Davis Technologies, Alachua, FL). We performed 60 min baseline recording in an open field, followed by a 60 min recording after a saline injection. *Gnal$^{+/-}$* and WT mice were then injected intraperitoneally with oxotremorine methiodide (0.1 mg/kg, Sigma-Aldrich), dissolved in saline (NaCl 0.9 g/l), and recorded another 60 min with the same protocol as for saline. Optogenetic stimuli were applied to the DN at low-frequency stimuli of 100 ms, 1.5 mW/mm$^2$, and 0.25 Hz. After 48–72 hr, a second session with θ-burst stimulations was performed. The mice were recorded 60 min after the saline injection and then 60 min after the oxotremorine M1 injection using the same protocol as in baseline day except that after 30 min two θ-burst sessions of 40 s each with 2 min pause in between were applied (a total of 600 pulses for each condition). The stimulations induced small twitches but no major motor effect.

## Histological verification of the site of optical fiber in DN and verification of the position of the electrodes

The animals were sacrificed with a single dose of pentobarbital (100 mg/kg, i.p.). Electrolytic lesions were performed to check the position of electrodes, mice were perfused with paraformaldehyde, and the brains were removed and kept in paraformaldehyde (4%). After slicing (using a vibratome at 90 µm thickness), all sites of the recordings were verified by superposing the atlas (Allen Brain Atlas) on slices, with the closest anatomical landmarks from our lesions used as reference points. Injections in the CL usually encompassed the neighboring structures (lateral mediodorsal thalamus, medial posterior thalamus), which have little if any projection to the dorsolateral striatum, but touched also the anterior part of the parafascicular thalamus that may contribute in part to the cerebello-thalamo-striatal pathway (*Xiao et al., 2018*).

## Behavioral analysis

Video recordings monitored the motor behavior of *Gnal$^{+/-}$* and WT mice in the open field. Dystonia severity was estimated using a previously published abnormal movement scoring scale (*Jinnah et al., 2000*; *Calderon et al., 2011*; *Pelosi et al., 2017*) for every 10-min-long blocks of the recording after the oxotremorine M injection was given. The assessment was blinded for mouse genotype and was done by two members of the team. The scale uses the following scores: 0 = normal motor behavior; 1 = no impairment, but slightly slowed movements; 2 = mild impairment: occasional abnormal postures and movements; ambulation with slow walk; 3 = moderate impairment: frequent abnormal postures and movements with limited ambulation; 4 = severe impairment: sustained abnormal postures without any ambulation or upright position. In addition, the total time of active wakefulness from the total time of recording (active wake percentages, AW%) was assessed for both states, pre- and post-oxotremorine and pre- and post θ-burst stimulation. Active wake was considered as the state when the mouse is exploring the open field by walking in any direction and is expressed as a percentage of the total time of the recording (*Georgescu et al., 2018*). We evaluated the impact of θ-burst DN stimulation on the onset of dystonic-like symptoms in saline- and oxotremorine-treated *Gnal$^{+/-}$* mice.

## Electrophysiological analysis

Spike sorting was completed using homemade MATLAB scripts (MathWorks, Natick, MA) based on *k-means* clustering on PCA of the spike waveforms (*Paz et al., 2006*). In order to evaluate the activity of the same cells in similar conditions during experiments, we investigated the change of the firing rate probability in the thalamus and M1 motor cortex during cerebellar DN 100 ms stimulations after saline or oxotremorine M administration was done and analyses in one continuous session. The average increase in firing rate during the stimulation was determined by computing the peristimulus time histogram (bin: 10 ms) of the spikes around the stimulation; the spike count in the histogram was divided by the duration of the stimulation and the number of stimulations administered to yield a firing rate. The acceleration of discharge due to the stimulation was taken as the average spike count during the stimulation subtracted by the baseline (taken as the 300 ms that preceded the stimulation onset). The response to stimulation was only analyzed in cells where at least one bin during the stimulation was four times larger than the standard deviation of the baseline values. To isolate in the

responses the part corresponding to the direct excitation (in the same neuron for DN recordings, after one synapse in the thalamus, and after two synapses in the cortex and striatum), we also measured the discharge in 10-ms-long windows: 0–10 ms after illumination onset in the DN, 4–14 ms after illumination onset in the thalamus, and 7–17 ms in the cortex.

## Statistics

Figures represent the averages ± standard error of the mean (SEM). We used nonparametric tests: Wilcoxon and Mann–Whitney statistics (depending on whether the measures were paired or unpaired). For factorial analysis, we used repeated-measures ANOVA. The statistical values were computed in Python using the modules SciPy (version 1.5.4), statsmodels (version 0.12.2), and scikit_posthocs (version 0.6.1). Boxplots are composed of a box that extends from the first quartile to the third quartile of the data, with a line at the median. The whiskers extend from the box by 1.5× the interquartile range, and data values falling outside the range of the whiskers are represented individually.

## Acknowledgements

This work was supported by the Agence Nationale de Recherche to DP and DH (ANR-16-CE37-0003 Amedyst), CL (ANR-17-CE37-0009 Mopla), the Fondation pour la Recherche Médicale (FRM-EQU202103012770), the Labex Memolife, and the Institut National de la Santé et de la Recherche Médicale (France). The authors declare no competing financial interests. The authors are very grateful to Sabine Meunier and Cecile Gallea for the critical reading of the manuscript.

## Additional information

### Funding

| Funder | Grant reference number | Author |
| --- | --- | --- |
| Agence Nationale de la Recherche | ANR-16-CE37-0003 Amedyst | Denis Hervé Daniela Popa |
| Agence Nationale de la Recherche | ANR-17-CE37-0009 Mopla | Clément Léna |
| Fondation pour la Recherche Médicale | FRM-EQU202103012770 | Clément Léna |
| Agence Nationale de la Recherche | ANR-19-CE37-0007 | Daniela Popa |
| Labex Memolife | | Clément Léna Daniela Popa |

The funders had no role in study design, data collection and interpretation, or the decision to submit the work for publication.

### Author contributions

Hind Baba Aïssa, Elena Laura Georgescu Margarint, Data curation, Formal analysis, Investigation, Writing – original draft; Romain W Sala, Data curation, Formal analysis, Investigation, Methodology, Software, Visualization, Writing – original draft; Jimena Laura Frontera, Investigation; Andrés Pablo Varani, Conceptualization, Data curation, Formal analysis; Fabien Menardy, Data curation; Assunta Pelosi, Resources; Denis Hervé, Funding acquisition, Resources; Clément Léna, Conceptualization, Formal analysis, Funding acquisition, Supervision, Validation, Visualization, Writing – original draft, Writing – review and editing; Daniela Popa, Conceptualization, Funding acquisition, Supervision, Writing – original draft, Writing – review and editing

### Author ORCIDs

Denis Hervé http://orcid.org/0000-0003-1376-1522
Clément Léna http://orcid.org/0000-0002-1431-7717
Daniela Popa http://orcid.org/0000-0002-8389-1122

## Ethics

Information on Gnal[+/-] mice are all provided in the Materials and Methods section. Ethics approval: APAFIS #29793-202102121752192 v3 & APAFIS #1334-2015070818367911.

### Decision letter and Author response

Decision letter https://doi.org/10.7554/eLife.79135.sa1
Author response https://doi.org/10.7554/eLife.79135.sa2

# Additional files

## Supplementary files

• Supplementary file 1. Tables of statistical results for all comparisons reported in the text.

• MDAR checklist

## Data availability

All data are available on Dryad repository https://doi.org/10.5061/dryad.p5hqbzkr9. The code for electrophysiological analysis is available on GitHub repository https://github.com/teamnbc/GNAL2022/, (copy archived at swh:1:rev:3ee1122cc220f91529a9d011d965e4ef4b72ed52).

The following dataset was generated:

| Author(s) | Year | Dataset title | Dataset URL | Database and Identifier |
|---|---|---|---|---|
| Baba Aïssa H, Sala R, Georgescu Margarint E, Frontera J, Varani A, Menardy F, Pelosi A, Herve D, Léna C, Popa D | 2022 | Electrophysiological and behavioral analysis of cerebello-cerebral coupling in GNAL+/- mice | https://dx.doi.org/10.5061/dryad.p5hqbzkr9 | Dryad Digital Repository, 10.5061/dryad.p5hqbzkr9 |

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
