## [Editor Report]

Baba Aissa et al. provide compelling evidence for a modulatory role of the cerebello-thalamo-striatal pathway in the pathology of DYT25 dystonia. Their results further suggest that cerebellar stimulation holds promise as a therapeutic intervention for treating dystonia.

---

## [Decision Letter]

**Decision letter after peer review:**

[Editors’ note: the authors submitted for reconsideration following the decision after peer review. What follows is the decision letter after the first round of review.]

Thank you for submitting your work entitled "Functional abnormalities in the cerebello-thalamic pathways in an animal model of dystonia" for consideration by *eLife*. Your article has been reviewed by 3 peer reviewers, and the evaluation has been overseen by a Reviewing Editor and a Senior Editor. The following individual involved in review of your submission has agreed to reveal their identity: Freek Hoebeek, PhD (Reviewer #1).

Our decision has been reached after consultation between the reviewers. Based on these discussions and the individual reviews below, we regret to inform you that your work will not be considered further at this time.

The Reviewers and Editors agreed that investigating a role for cerebello-thalamic function within the context of dystonia was interesting and exciting, and they all appreciated use of optogenetic approaches in a mouse model of dystonia to begin to address this question in a mechanistic way. Unfortunately, the Reviewers agreed that substantial additional experiments would be required. These were expected to take more than a few months, with uncertain outcome. Therefore, we are unable to proceed toward publication. However, due to the overall enthusiasm for the work that was shared by the Reviewers and Editors, should you be able to address the concerns raised by the reviewers, we are open to the possibility of receiving a new submission in the future.

The points of concern were:

1. Further analysis/ verification to tighten the links between the LCN-CL^-^striatum was requested by all three reviewers and was seen as fundamental.

2. Further analysis of apparent gender differences was requested by two reviewers.

3. The relatively small effect of the theta burst stimulation tempered the enthusiasm for its putative therapeutic relevance.

4. There were issues with writing and presentation that were raised by all three Reviewers. It was generally thought that a tightening of the language may help clarify some of the missing mechanistic links.

*Reviewer #1:*

The authors investigated how the cerebellar output from lateral nuclei affects neuronal firing in the ventrolateral (VL) and centrolateral (CL) nuclei of the thalamus and the primary motor cortex (M1) in wildtype and Gnal+/- mice. Using standard optogenetics the authors stimulate neurons in the lateral cerebellar nuclei (LCN) with theta frequency and record the changes in electrophysiology and behavior. They do so before and after the administration of oxotremorine, which in Gnal+/- induces dystonic behavior. Since systemic administration of this muscarinic cholinergic agonist, as well as its application to the basal ganglia but not to the cerebellum, induces dystonic events, the authors investigated whether cerebellar manipulation could influence the thalamo-cortical motor system.

Although this manuscript taps into a research questions that is of importance for a broad scope of neuroscience research fields (cerebellar, thalamo-cortical, striatal, dystonic, developmental disorders, etc), in its current form this manuscript is in my opinion not ready for publication, because mechanistically the gain of the current manuscript seems limited. However, with the standardized methodology more insights can be provided on the biological principles underlying the cerebellar impact on the other parts of the dystonic motor system.

My comments all concern the need for more recordings or anatomical analysis:

1) Throughout the dystonia research a lot of emphasis has been put on the striatum – this forms the basis of the current paper. However, even though (in my opinion) technically possible, the authors did not record from the striatum. This makes that one of the primary output regions of the CL thalamus is not investigated. Yet, the authors did record in the primary output region of the VL, being the primary motor cortex (M1). The manuscript would gain a lot of insight if the comparison between the major outputs of VL and CL (M1 and striatum, respectively) would be investigated. Moreover, such additional recordings could also provide some clues as to why the difference between WT and Gnal+/- in responses to LCN stimulation before vs after theta stimulation is not significant, whereas this is significant for VL and M1 (Figure 3D). I would have expected that the plasticity of the cerebello-thalamic circuit is particularly effective on the most direct connection between the cerebellum, being the CL^-^->striatum tract.

2) The authors report the effect of theta stimulation of LCN on thalamic and M1 spiking patterns as well as behavior, but not on cerebellar spiking patterns. This is of major importance, since it may be that the LCN spiking pattern is altered by local theta stimulation. Any change in cerebellar spiking could have an effect on thalamic and cortical activity, as well as on behavior. In a similar fashion the authors should investigate the impact of oxotremorine on spiking patterns of LCN in their experimental setting.

3) The authors show that the responses in VL and CL thalamic neurons upon 2 trains of theta frequency stimulation in the LCN stimulation are different. In a rather isolated report, Aumann et al., 2000 (Neurosci lett) reported that high frequency stimulation of cerebello-thalamic axons induced long-term potentiation of these synapses. A histological examination of the tissue of the current study could also provide clues as to whether the potentiation and depression have a synaptic base or are rather intrinsic. Moreover, these additional readouts provide valuable clues about whether the alterations in cerebello-thalamic functioning may be partially due to structural alterations. Hereto the authors could choose to use the tissue transfected with AAV2.1, or use classical neurochemical tracers (such as used by Teune et al., 2000, Brain Res).

*Reviewer #2:*

Margarint et al. provide an interesting study of the role of cerebello-thalamic projections on modulating a striatal mouse model of dystonia. Understanding how key motor areas, such as the basal ganglia and cerebellum, interact to produce disease-related motor dysfunction is an important area of inquiry and the present study addresses the problem head-on. There are several areas, however, by which the study could be improved. A major concern is that the language used in the manuscript is unclear in many instances and obfuscates the findings and the experimental design. In addition, there is an interchangeable use of qualitative (mild, strong, major etc) and quantitative (significant) descriptors of the included results which sometimes makes evaluation of the data difficult, particularly as one is reading through the Results section.

Overall, it is not entirely clear from the introduction what the main motivation of the study actually is. However, later on in the Discussion, the authors state "If cerebellar dysfunction may disrupt basal ganglia function, the reverse has received little attention so far". I would suggest moving this to the introduction, and further building out the idea from the outset.

From a technical perspective, there needs to be a resolution of the male and female difference seen in the behavior and whether that difference is meaningful to the physiology data that are reported. The starting point may be to increase the (n) in Figure 1 experiments. If the motor differences persist in an adequately powered experiment, then having sufficient cells to evaluate the electrophysiologic differences between males and females would be a necessary step to undertake.

The authors ought to include histology verifying the DN-VAL/CL projections that are assumed to underlie the optogenetic modification of the oxotremorine phenotype in the Gnal animals. (please see specific comments for further details).

152-162: Given the extensive connectivity of the cerebellar nuclei to subcortical structures, histology showing whether synapses from AAV injected neurons are present in VAL and CL would be important to include.

Figure 2C: Please provide much higher power images for the lesion sites.

*Reviewer #3:*

Authors presented an interesting paper on the cerebello-thalamal dysfunction in an animal model of dystonia (DYT25) and theta burst stimulation at the cerebellar nuclei could alleviate the dystonic symptoms, which might have potential clinical implications. While this manuscript is interesting, there are numbers of concerns:

1. There are gender differences in disease phenotype at baseline (Figure 1); however, it is not clear whether the gender difference also affects cerebello-thalamal physiology, responses to oxotremorine, and responses to cerebellar stimulation. Gender differences may need to be investigated throughout the manuscript.

2. Thalamic CL has the most obvious physiological changes in both WT and mutant mouse line upon oxotremorine administration. However, cerebellar stimulation is at the dentate nucleus. Does dentate nucleus have direct connectivity to CL?

3. As author mentioned, oxotremorine mostly acts upon striatum to induce dystonia in this mouse model based on the previous work, how does the altered striatal activity drive cerebello-thalamal abnormal activity to cause dystonia? The hypothesis needs to be clearly stated and directly tested. And what is the structural and expression pattern of GNAL throughout the brain. Is there any DCN or CL expression?

4. The main clinical implication for the manuscript is theta burst stimulation can reduce dystonic movement (Figure 4B). However, the degree of reduction is very minor (perhaps 10% only). Therefore, the clinical utility is in question and whether this could be used in the future clinical trial is unknown.

[Editors’ note: further revisions were suggested prior to acceptance, as described below.]

Thank you for resubmitting your work entitled "Functional abnormalities in the cerebello-thalamic pathways in a mouse model of DYT25 dystonia" for further consideration by *eLife*. Your revised article has been evaluated by Timothy Behrens (Senior Editor) and a Reviewing Editor.

The manuscript has been improved but there are some remaining issues raised by Reviewer 2 that need to be addressed, please see below.

*Reviewer #1 (Recommendations for the authors):*

The authors addressed my comments and I have no further critiques.

*Reviewer #2 (Recommendations for the authors):*

My previous comments have been adequately addressed. Some remaining thoughts:

Although the authors discuss the importance of the regularity of cerebellar firing patterns in dystonia models, they do not report any value on the regularity of thalamic firing patterns (solely on firing rate). Please consider adding some supplementary representation of the regularity of thalamic firing in WT and Gna+/- mice.

In supplementary Table 3 there seems to be something wrong with the p-values. Please check.

In the first sentence of the second paragraph of the introduction, it is not clear what type of task is meant with 'temporal discrimination threshold'. Please elaborate.

---

## [Author Response]

[Editors’ note: the authors resubmitted a revised version of the paper for consideration. What follows is the authors’ response to the first round of review.]

The points of concern were:1. Further analysis/ verification to tighten the links between the LCN-CL^-^striatum was requested by all three reviewers and was seen as fundamental.

We have performed new experiments in order to address these requests. We have obtained specific infection of the DN-CL pathway by using a retrograde AAV expressing ChR2 injected in CL. (ADdgene pAAV-hSyn-hChR2(H134R)-EYFP ref 26973). We then proceeded to record extracellular activity from DN and dorsolateral striatum (DLS) during optogenetic stimulations of the DN. First, the presence of infected neurons in the contralateral DN from retrograde injections from CL demonstrated that anatomical projections exist from the deep cerebellar nuclei to the CL. Following stimulations of the DN, we observed short latency excitatory responses in DLS coherent with a di-synaptic projection from the dentate nucleus. We not only observed these responses but also examined how they were changed by theta-burst stimulations. Interestingly, we found that the response in fast-spiking neurons was primarily affected. These new results are presented in Figure 4.

2. Further analysis of apparent gender differences was requested by two reviewers.

We performed additional behavioral experiments in order to evaluate possible gender differences in WT or Gnal+/- mice. The results of the tests (Vertical pole test, Horizontal bar test, Grid Test, Fixed speed rotarod, Gait test, Average speed, and distance travelled during an openfield session) are shown in Figure 1. Overall, both genders performed similarly on the different motor assessments, in both genotypes. We found one exception for the horizontal bar, where WT females performed better than males. We postulate that in our hands, the horizontal bar test might involve other components affecting the motor performance, such as anxiety or motivation, but could not impute our differences directly to gender due to the difference only being present in WT females.

3. The relatively small effect of the theta burst stimulation tempered the enthusiasm for its putative therapeutic relevance.

We agree with the reviewer that the amplitude of the effect remains modest. Yet, the scale of improvement in most intervention in patients (Oyama and Hattori, 2021) remains in the same order of magnitude (except for tremor symptoms), and cerebellar theta stimulations using rTMS result in a clinical improvement of 15% of dystonia symptoms (Koch et al., 2014; Bradnam et al., 2016), which may improve the quality of life score in comparable amounts with botulinium toxin injections (Bradnam et al., 2016). Finally, the benefit was observed after a single session in our model, while in patients this may not be sufficient to produce beneficial effect (Linssen et al., 2015; Meunier et al., 2015). However, in our model, dystonia is induced by pharmacological injections which also have side effects that might build-up upon repeated sessions. Using old (>1 yr) animals (which develop spontaneous dystonia symptoms but are harder to get and could thus not be used in our study) might be more relevant to test the effect of repeated stimulation sessions.

4. There were issues with writing and presentation that were raised by all three Reviewers. It was generally thought that a tightening of the language may help clarify some of the missing mechanistic links.

We extensively revised the whole manuscript for improved clarity. In particular, we now refer to the altered cerebello-thalamic excitability as a potential endophenotype.

Reviewer #1:The authors investigated how the cerebellar output from lateral nuclei affects neuronal firing in the ventrolateral (VL) and centrolateral (CL) nuclei of the thalamus and the primary motor cortex (M1) in wildtype and Gnal+/- mice. Using standard optogenetics the authors stimulate neurons in the lateral cerebellar nuclei (LCN) with theta frequency and record the changes in electrophysiology and behavior. They do so before and after the administration of oxotremorine, which in Gnal+/- induces dystonic behavior. Since systemic administration of this muscarinic cholinergic agonist, as well as its application to the basal ganglia but not to the cerebellum, induces dystonic events, the authors investigated whether cerebellar manipulation could influence the thalamo-cortical motor system.Although this manuscript taps into a research questions that is of importance for a broad scope of neuroscience research fields (cerebellar, thalamo-cortical, striatal, dystonic, developmental disorders, etc), in its current form this manuscript is in my opinion not ready for publication, because mechanistically the gain of the current manuscript seems limited. However, with the standardized methodology more insights can be provided on the biological principles underlying the cerebellar impact on the other parts of the dystonic motor system.My comments all concern the need for more recordings or anatomical analysis:1) Throughout the dystonia research a lot of emphasis has been put on the striatum – this forms the basis of the current paper. However, even though (in my opinion) technically possible, the authors did not record from the striatum. This makes that one of the primary output regions of the CL thalamus is not investigated. Yet, the authors did record in the primary output region of the VL, being the primary motor cortex (M1). The manuscript would gain a lot of insight if the comparison between the major outputs of VL and CL (M1 and striatum, respectively) would be investigated. Moreover, such additional recordings could also provide some clues as to why the difference between WT and Gnal+/- in responses to LCN stimulation before vs after theta stimulation is not significant, whereas this is significant for VL and M1 (Figure 3D). I would have expected that the plasticity of the cerebello-thalamic circuit is particularly effective on the most direct connection between the cerebellum, being the CL^-^->striatum tract.

The CL remains a structure difficult to target in vivo due to its small mediolateral extension in the mouse. Following the reviewer’s suggestion, we have further investigated the DN-CL^-^Striatum pathway, we directly targeted the DN-CL pathway by optogenetic stimulations of DN neurons retrogradely infected from the CL area, and recording the neuronal responses in the striatum. We found indeed in these experiments some cells in the striatum which are excited by the DN with a short latency, although the increase in discharge remained modest. Then we investigated the impact of theta-burst stimulation and found that it differentially evoked responses changes in the striatum (mostly for fast spiking striatal neurons) in WT vs Gnal+/- mice. Overall, the changes in the striatal activity evoked by theta-burst stimulations are congruent with those found in the CL. These results are presented in a novel Figure 4. We feel however that it remains difficult to compare the amplitude of plasticity in the DN-CL^-^striatum vs DN-VAL-M1, given the signal/noise ratio of its measure.

2) The authors report the effect of theta stimulation of LCN on thalamic and M1 spiking patterns as well as behavior, but not on cerebellar spiking patterns. This is of major importance, since it may be that the LCN spiking pattern is altered by local theta stimulation. Any change in cerebellar spiking could have an effect on thalamic and cortical activity, as well as on behavior. In a similar fashion the authors should investigate the impact of oxotremorine on spiking patterns of LCN in their experimental setting.

We performed theta-burst optogenetic stimulations of the DN and recorded from those cells. We observe during theta-burst stimulations in WT mice, an entrainment of DN neurons to the frequency of stimulation, but a return to a baseline FR at the end of the theta protocol. Hence, the theta burst stimulations do not induce a lasting change in the firing pattern of DN neurons nor change in response to the optogenetic stimulations. These results are presented in figure 4B-D and Supplementary Figure 3.

Oxotremorine produces a drop in DN neurons firing rate in both WT and Gnal+/-

(Supplementary Figure 4). It is likely that this drop in firing rate in DN neurons is partially due to the diminution of locomotor activity induced by the cholinergic shock. Notably, we did not observe a strong increase in DN firing irregularity; this differs substantially from other rodent model of dystonia possibly since these other models have an induction of the symptoms caused by cerebellar anomalies, while in Gnal+/- mice, oxotremorine injection in the striatum suffices to trigger dystonia. This is discussed in the Discussion section.

3) The authors show that the responses in VL and CL thalamic neurons upon 2 trains of theta frequency stimulation in the LCN stimulation are different. In a rather isolated report, Aumann et al., 2000 (Neurosci lett) reported that high frequency stimulation of cerebello-thalamic axons induced long-term potentiation of these synapses. A histological examination of the tissue of the current study could also provide clues as to whether the potentiation and depression have a synaptic base or are rather intrinsic. Moreover, these additional readouts provide valuable clues about whether the alterations in cerebello-thalamic functioning may be partially due to structural alterations. Hereto the authors could choose to use the tissue transfected with AAV2.1, or use classical neurochemical tracers (such as used by Teune et al., 2000, Brain Res).

We thank the reviewer for this suggestion. Unfortunately, our approach failed to provide an accurate assessment of the density or size of the axonal varicosities, which would document the anatomical counterpart to our findings. We have added a reference to Aumann’s work in the text. We agree that our study prompts further examination of the plasticity since we could not more determine whether the potentiation results from an increased cerebello-thalamic synaptic transmission or an increased excitability of the post-synaptic thalamic neurons.

Reviewer #2:Margarint et al. provide an interesting study of the role of cerebello-thalamic projections on modulating a striatal mouse model of dystonia. Understanding how key motor areas, such as the basal ganglia and cerebellum, interact to produce disease-related motor dysfunction is an important area of inquiry and the present study addresses the problem head-on. There are several areas, however, by which the study could be improved. A major concern is that the language used in the manuscript is unclear in many instances and obfuscates the findings and the experimental design. In addition, there is an interchangeable use of qualitative (mild, strong, major etc) and quantitative (significant) descriptors of the included results which sometimes makes evaluation of the data difficult, particularly as one is reading through the Results section.Overall, it is not entirely clear from the introduction what the main motivation of the study actually is. However, later on in the Discussion, the authors state "If cerebellar dysfunction may disrupt basal ganglia function, the reverse has received little attention so far". I would suggest moving this to the introduction, and further building out the idea from the outset.

We thank the reviewer for raising our attention on these multiple problems. The manuscript has therefore been extensively rewritten.

From a technical perspective, there needs to be a resolution of the male and female difference seen in the behavior and whether that difference is meaningful to the physiology data that are reported. The starting point may be to increase the (n) in Figure 1 experiments. If the motor differences persist in an adequately powered experiment, then having sufficient cells to evaluate the electrophysiologic differences between males and females would be a necessary step to undertake.

Additional experiments allowed us to verify that, with a sufficient number of mice, male and female do not exhibit difference on behavioral tests, except for the horizontal bar test, but even in that case, the mutation of GNAL did not affect the behavior of males or females. There was therefore no indication of a differential sensitivity to the GNAL mutation between genders.

The authors ought to include histology verifying the DN-VAL/CL projections that are assumed to underlie the optogenetic modification of the oxotremorine phenotype in the Gnal animals. (please see specific comments for further details).

This is addressed in the new figure 4.

152-162: Given the extensive connectivity of the cerebellar nuclei to subcortical structures, histology showing whether synapses from AAV injected neurons are present in VAL and CL would be important to include.

The presence of synapses in VAL and CL has been widely documented (a few references have been added, notably (Gornati et al., 2018)).

Figure 2C: Please provide much higher power images for the lesion sites.

Histological images are provided in Supplementary figure 1.

Reviewer #3:Authors presented an interesting paper on the cerebello-thalamal dysfunction in an animal model of dystonia (DYT25) and theta burst stimulation at the cerebellar nuclei could alleviate the dystonic symptoms, which might have potential clinical implications. While this manuscript is interesting, there are numbers of concerns:1. There are gender differences in disease phenotype at baseline (Figure 1); however, it is not clear whether the gender difference also affects cerebello-thalamal physiology, responses to oxotremorine, and responses to cerebellar stimulation. Gender differences may need to be investigated throughout the manuscript.

As noted by another reviewer, the number of male/female mice used on the previous version of the manuscript was limited; this has prompted a new series of animals which showed no differential sensitivity of male and female Gnal+/- mice to the mutation.

2. Thalamic CL has the most obvious physiological changes in both WT and mutant mouse line upon oxotremorine administration. However, cerebellar stimulation is at the dentate nucleus. Does dentate nucleus have direct connectivity to CL?

The anatomical and physiological direct connections have been demonstrated previously (Ichinoe et al. 2000, Chen 2014). Furthermore, in a new set of experiments we combined retrograde infections from the CL to express ChR2 in the DN, with illumination of the DN and recording in the dorso-lateral striatum. We observed short-latency responses in some DLS neurons, consistent with functional di-synaptic DN-CL^-^DLS pathway. This is more clearly explained in the text and references to the existing literature have been completed.

3. As author mentioned, oxotremorine mostly acts upon striatum to induce dystonia in this mouse model based on the previous work, how does the altered striatal activity drive cerebello-thalamal abnormal activity to cause dystonia? The hypothesis needs to be clearly stated and directly tested. And what is the structural and expression pattern of GNAL throughout the brain. Is there any DCN or CL expression?

We have rewritten the introduction to improve the motivation of our study. Reference to the characterization of the GNAL expression has been added (Belluscio et al., 1998; Vemula et al., 2013). GNAL expression is strongly expressed in the striatum (and involved in striatal transmission). It is also expressed in the cerebellar Purkinje cell, but we are not aware of studies demonstrating its function in these cells. In several rodent models of dystonia, the cerebellar nuclei neurons express an anomalously irregular firing in the cerebellar nuclei which has been proposed to cause the dystonic attack via a transmission to the striatum (Chen et al. 2014). We added recordings in the DN, which do not show increased irregularity of the DN neurons in Gnal+/- mice. Therefore, we do not believe that the cerebellum causes the dystonic movements, but instead that the changes in the cerebello-thalamic excitability reflects maladaptive reaction to the anomalies of the basal-ganglia output.

4. The main clinical implication for the manuscript is theta burst stimulation can reduce dystonic movement (Figure 4B). However, the degree of reduction is very minor (perhaps 10% only). Therefore, the clinical utility is in question and whether this could be used in the future clinical trial is unknown.

We agree that the amplitude of the effect is moderate; yet similar values have been obtained in patients and still represent an improvement of life quality. Moreover, we only used a single session of stimulations, and repeated treatments have been shown in patients to increase the size effect (although it is still unclear if the mechanisms involved in our DN stimulations and in clinical transcranial stimulations are similar). This is now discussed in the Discussion.

[Editors’ note: what follows is the authors’ response to the second round of review.]

The manuscript has been improved but there are some remaining issues raised by Reviewer 2 that need to be addressed, please see below.Reviewer #2 (Recommendations for the authors):My previous comments have been adequately addressed. Some remaining thoughts:Although the authors discuss the importance of the regularity of cerebellar firing patterns in dystonia models, they do not report any value on the regularity of thalamic firing patterns (solely on firing rate). Please consider adding some supplementary representation of the regularity of thalamic firing in WT and Gna+/- mice.

We added a supplementary figure (Supplementary Figure 3, Supplementary Tables 15,16) with the representation of the regularity of thalamic firing in WT and Gnal+/- mice. We also report and discuss these results in the text.

In supplementary Table 3 there seems to be something wrong with the p-values. Please check.

Thank you. We checked and corrected the p-values in the Supplementary Tables.

In the first sentence of the second paragraph of the introduction, it is not clear what type of task is meant with 'temporal discrimination threshold'. Please elaborate.

Thank you. We described more in detail what means 'temporal discrimination threshold'.